# Better Outcomes with Intranigral versus Intrastriatal Cell Transplantation: Relevance for Parkinson’s Disease

**DOI:** 10.3390/cells11071191

**Published:** 2022-04-01

**Authors:** Marine Droguerre, Sébastien Brot, Clément Vitrac, Marianne Benoit-Marand, Laure Belnoue, Maelig Patrigeon, Anaïs Lainé, Emile Béré, Mohamed Jaber, Afsaneh Gaillard

**Affiliations:** 1Laboratoire de Neurosciences Expérimentales et Cliniques, Inserm, Université de Poitiers, 86022 Poitiers, France; marine.droguerre@gmail.com (M.D.); sebastien.brot@univ-poitiers.fr (S.B.); clement.vitrac@uzh.ch (C.V.); marianne.benoit.marand@univ-poitiers.fr (M.B.-M.); laure.belnoue@univ-poitiers.fr (L.B.); maelig.patrigeon@univ-poitiers.fr (M.P.); anais.laine@univ-poitiers.fr (A.L.); emile.bere@univ-poitiers.fr (E.B.); mohamed.jaber@univ-poitiers.fr (M.J.); 2CHU de Poitiers, 86022 Poitiers, France

**Keywords:** Parkinson’s disease, intranigral transplantation, L-DOPA, behavior, electrophysiological assessment

## Abstract

Intrastriatal embryonic ventral mesencephalon grafts have been shown to integrate, survive, and reinnervate the host striatum in clinical settings and in animal models of Parkinson’s disease. However, this ectopic location does not restore the physiological loops of the nigrostriatal pathway and promotes only moderate behavioral benefits. Here, we performed a direct comparison of the potential benefits of intranigral versus intrastriatal grafts in animal models of Parkinson’s disease. We report that intranigral grafts promoted better survival of dopaminergic neurons and that only intranigral grafts induced recovery of fine motor skills and normalized cortico-striatal responses. The increase in the number of toxic activated glial cells in host tissue surrounding the intrastriatal graft, as well as within the graft, may be one of the causes of the increased cell death observed in the intrastriatal graft. Homotopic localization of the graft and the subsequent physiological cell rewiring of the basal ganglia may be a key factor in successful and beneficial cell transplantation procedures.

## 1. Introduction

Parkinson’s disease (PD) is the most common progressive neurodegenerative movement disorder, characterized mainly, but not exclusively, by a marked ongoing degeneration of mesencephalic dopaminergic neurons in the substantia nigra pars compacta (SNpc), leading to a loss of dopamine axon terminals and a reduction in dopamine in the target structure, the striatum [1]. Clinically, this dopaminergic deficit is characterized by motor disabilities, such as rigidity, resting tremor, altered gait, and postural instability [2,3]. Many therapeutic approaches have been deployed to manage PD symptoms, such as pharmacological L-3,4-dihydroxyphénylalanine (L-DOPA) treatments [4] or deep brain stimulation [5]. However, neither of these approaches stops or reverses the progress of dopamine neuron degeneration in the SNpc, nor do they restore the physiological basal ganglia loops. An alternative strategy for managing PD is the transplantation of fetal tissue from the ventral mesencephalon (VM) and, more recently, the induction of pluripotent stem- (iPS) derived dopamine neurons to replace lost dopamine [6,7,8,9]. In the vast majority of transplantation experiments, dopaminergic cells have been placed not in their ontogenic site, the SNpc, but in their target area, the striatum [10,11,12,13,14]. Nevertheless, these ectopically transplanted cells have been shown to survive [15], produce, and secrete dopamine within the host striatum [15,16,17]; they have attenuated drug-induced rotational asymmetry [16,18] and have established synaptic connections with host striatal neurons [15,19,20,21,22,23,24]. Moreover, clinical trials of intrastriatal neural transplantation have shown robust reinnervation of the striatum, as seen with positron emission tomography [11,13,25,26] and postmortem studies [27]. Synaptic dopamine release for as long as 10 years after transplantation was also documented [14]. Three concomitant studies have reported a decade ago that intrastriatal grafts can survive for up to 14 years and produce significant but incomplete benefits to patients [28,29,30]. One of the critical variables that influences the efficacy of clinical neural transplantation is optimal graft placement [31,32]. Intrastriatal transplantation of VM tissue does not allow the reconstruction of the nigrostriatal pathway, excludes the grafts from specific afferent inputs, and does not restore the basal ganglia loops. The main reason for this ectopic graft placement is the assumed inability of dopaminergic neuroblasts placed in the adult substantia nigra to extend axons over long distances to reach their target area within the striatum [33,34]. Several studies have today refuted this misconception [35,36,37,38,39]. In previous studies, others and we have shown that embryonic VM tissue obtained from green fluorescent protein (GFP) mouse fetuses and grafted into the 6-Hydroxydopamine- (6-OHDA) lesioned SNpc of adult mice can survive, differentiate into dopamine neurons, and develop substantial projections through the medial forebrain bundle (mfb) to the striatum [37,39]. Grafted cells expressed dopaminergic markers, and most of the newly developed dopaminergic neurons co-expressed the substantia nigra marker G-protein-coupled inwardly rectifying potassium channels (Girk2), while a lesser proportion expressed the ventral tegmental area (VTA) marker calbindin. Dopaminergic nigrostriatal circuit reconstruction has been accompanied by an increase in striatal dopamine levels and a complete abolishment of apomorphine-induced rotations as compared to unilaterally pre-lesioned animals prior to transplantation [37,39]. While previous studies have reported that both intranigral and intrastriatal transplants of VM tissue were able to induce amelioration of simple motor behavior, such as drug-induced rotations [39,40,41,42,43], only a few studies have documented whether any of these grafts have been able to induce recovery of more complex motor behavior [42,44,45]. More generally, it is still not clear to date to what extent intranigral grafts can potentially be more effective than intrastriatal ones. This is of major importance in cell transplantation strategies, and answering this question may provide a change of concept and potential significant benefits to PD patients who would undergo this procedure.

In this study, we compared side-by-side the anatomical and functional effects of fetal VM grafts placed in the striatum or the SNpc of 6-OHDA-lesioned mice. The functional effects were tested using a battery of simple to complex behavioral tests and electrophysiological measurements, while the neuroanatomical integration of transplanted cells was evaluated by immunohistochemical analysis.

## 2. Materials and Methods

### 2.1. Animals

Animal experimentation and housing were carried out in accordance with the guidelines of the French Agriculture and Forestry Ministry (decree 87849) and the European Communities Council Directive (2010/63/EU). The procedures referenced under the file number APAFIS#4928-20 16041 117503028 v3 were approved by ethics committee N°84 COMETHEA Poitou-Charentes. All mice were housed under a standard 12 h light: dark cycle. The animals had ad libitum access to food and water. All efforts were made to reduce the number of animals used and their suffering. A total of 118 animals were used in this study (Table 1). During the staircase test training period, 6 mice failing to reach a criterion of at least 2 pellets collected with the paw were eliminated from the study. Of the remaining 112, 22 remained intact, and 90 animals were lesioned, among which 33 were transplanted into the lesioned SNpc (intranigral transplant) and 29 into the striatum (intrastriatal transplant). Among the 62 transplanted mice, 6 were transplanted with beta-actin-GFP mouse fetal VM tissue (*n* = 3 intrastriatal; *n* = 3 intranigral) and 56 with tyrosine hydroxylase-GFP (TH-GFP) mouse fetal VM tissue (*n* = 26 intrastriatal; *n* = 30 intranigral). The 6 mice transplanted with VM fetal tissue from beta-actin-GFP mice were used for immunohistological analysis of the transplant. A total of 39 mice were used for behavioral tests (*n* = 8 control; *n* = 9 lesioned; *n* = 11 intrastriatal; *n* = 11 intranigral). A total of 31 mice were used for amphetamine-induced rotation to evaluate the effects of removal of the intranigral graft (*n* = 12 lesioned; *n* = 8 intrastriatal; *n* = 5 intranigral; *n* = 6 intranigral-removed). In total, 37 mice were used for the electrophysiological studies (*n* = 14 control; *n* = 8 lesioned; *n* = 8 intrastriatal; *n* = 8 intranigral). An amount of 1 mouse from each transplanted group (*n* = 1 intrastriatal; *n* = 1 intranigral) was used for electron microscopy.

### 2.2. Surgical Procedures

6-OHDA Lesioning: Adult female C57BL/6 mice (4–6 months old, Janvier Labs, Le Genest-Saint Isles, France) weighing 20–25 g at the time of surgery were used in this study. Under ketamine-xylazine anesthesia (intraperitoneal, 100 mg/kg and 10 mg/kg body weight, respectively), mice received into the left SNpc 1 μL 6-OHDA (8 μg/μL in 0.2 mg/mL l-ascorbate-saline; Sigma-Aldrich, Lyon, France) at the following coordinates (in mm relative to bregma): AP, −3.1; ML, 1.4; DV, 3.8. The drug was injected slowly (0.5 µL/min), and the cannula was left in place for 5 min to allow diffusion of the solution before being retracted.

*Transplantation*: To facilitate the identification of the transplanted dopamine neurons and their axonal projections within the host brain, donor VM tissue cells were isolated from transgenic mice overexpressing GFP under the influence of the TH promoter [46]. To analyze the cellular composition of the grafts, VM donor tissues were isolated from transgenic mice over-expressing the GFP under the influence of the beta-actin promoter [47]. Transplantation was performed according to a previously described cell suspension technique [37,48]. Timed pregnant mice (the day of vaginal plug was designated embryonic day 0.5 (E0.5)) were generated by crossing wild-type C57BL/6 females with heterozygous transgenic C57BL/6 males expressing either GFP under the influence of the 9-kb upstream region of the rat TH gene or GFP under chicken beta-actin promoter. VM tissues were removed from the embryos at E12.5 and collected in a basic medium (0.6% glucose in saline). After removing dura and vessels, the tissues were chopped into small pieces and then dissociated into cell suspensions by gentle trituration using a fire-polished Pasteur pipette. The viability of cells was established before transplantation by a Trypan blue dye exclusion test and was found to be 95%. Three weeks after 6-OHDA lesioning, 1.5 µL containing approximately 150,000 cells (100,000 cells/μL) was injected unilaterally into either the substantia nigra or the striatum of 6-OHDA-lesioned mice at the following stereotaxic coordinates (respectively: AP, −3.1; ML, 1.3; DV, 3.7 and AP, 0.5; ML, 2; DV, 2.8) using a 5 μL Hamilton syringe. The cells were stored on ice and gently triturated to maintain homogeneous suspension during the entire transplantation procedure. For each set of experiments, the same cell preparation, the same number, and the same volume of cells was used to graft intranigral and intrastriatal animals. During the grafting procedure, we alternated the animals by grafting one mouse in the striatum and one mouse in the substantia nigra, and so on.

### 2.3. Behavioral Tests

Cylinder: Forelimb use asymmetry during vertical exploration was assessed using the cylinder test [49]. The mice were tested for paw preference in a spherical glass beaker (ø 11.5 cm) during a 3 min session before lesion surgery [50] and then three and ten weeks after transplantation. Two cameras were positioned in order to ensure a 360° view to the observer. The session was videotaped and later scored by detailed freeze frame analysis. Forelimb touches were counted, and simultaneous paw touches were excluded from the analysis. Two lesioned mice and one intranigral-grafted mouse that failed to reach 20 touches during the baseline session were excluded from the analysis [51,52]. Data were expressed as the percentage of contralateral touches calculated as (contralateral touches)/(ipsilateral touches + contralateral touches) × 100 [53,54].

Challenging beam traversal: Motor performance was measured 6 weeks post-transplantation using a beam-walking test [55,56] with minor modifications. The beam consisted of four sections (25 cm each, 1 m total length) of Plexiglas. The animals were trained to traverse the length of the beam from the widest section (3.5 cm) to the narrowest (0.5 cm), which directly led into their home cage. At the time of the test, a wire mesh cover (1 cm^2^) of corresponding width was placed on the beam surface to increase the difficulty in such a way that, in a forward motion of the forelimb, the paw could be “placed” on the grid or “slipped” through the grid. Two days before testing, the animals were trained without the grid with two assisted trials and were able to traverse the beam until they completed five unassisted runs across the entire length of the beam. On the second day of training, the animals were allowed to run five trials. Prior to starting the test, the mice were allowed to accomplish two trials on the beam and two trials with the grid. The mice were then videotaped while traversing the grid-surfaced beam for a total of three trials. The videotapes were viewed and rated in slow motion for measuring time to traverse, and for each forelimb, the percentage of foot slips was calculated according to the following formula: % of foot slips (contralateral or ipsilateral) = number of foot slips (contralateral or ipsilateral)/(number of foot placed (contralateral or ipsilateral) + number of foot slips (contralateral or ipsilateral)) × 100.

Apomorphine- and amphetamine- induced rotations: Apomorphine- and amphetamine -induced rotational behaviors were assessed using an automatic rotameter (Omnitech Electronics, Colombus, OH, USA) according to Ungerstedt and Arbuthnott (1970) [57]. Full body turns were monitored over a period of 60 min following an injection of apomorphine (0.5 mg/kg, subcutaneously) or d-amphetamine sulphate (5 mg/kg, intraperitoneally (i.p.)) [54,58]. The animals were tested 3 weeks after lesion surgery and 13 weeks after grafting in order to measure dopaminergic imbalance. Data were expressed as net rotations (total right—total left 360° turns) per 60 min.

Staircase: The staircase test was designed for the objective assessment of independent forelimb use in skilled reaching and grasping abilities [59,60]. The mice were food-restricted for 4 days prior to testing, while maintaining 90% of their body weight. Prior to lesion surgery, the mice were trained to collect pellets for 15 min daily over three weeks and then for 10 min once a week during three consecutive weeks to establish baseline performance. The animals were tested during six consecutive weeks post-grafting after two resting weeks. In addition, the mice were tested with the staircase test during two consecutive days four months after transplantation in order to test the effect of a single injection of L-DOPA on their grasping performance. The sessions were videotaped by placing a camera on each side of the staircase, allowing the scoring of both the number of pellets collected by each paw individually and the trajectory of the grasping movement. Pellets that were dropped or collected with the tongue were excluded from the analysis. Reaching deficits were expressed in terms of contralateral side bias, i.e., the number of pellets retrieved on the right side (impaired, contralateral to lesion) as a percentage of the total number retrieved on each side.

Videotaping and kinematics: Video recordings were conducted with a miniature monochrome camera (ViewPoint, Lyon, France) positioned on the right side of the apparatus. The camera was set to 25 frames/s with a resolution of 500 × 582. Each right grasping movement was divided frame-by-frame, captured, and saved using an ethological keyboard (Labwatcher, View Point). A movement component rating scale derived from Eshkol–Wachman Movement Notation (EWMN) allowed us to decompose and divide the entire grasping movement into three task phases: the reaching movement ranging from the moment the paw was lifted from the ground until it touched the pellet; the grasping movement corresponding to the closing of the digits; and the retrieval movement up to the release of the pellet into the mouth. To measure the time required for each task phase, we selected right successful reaches (the mouse retrieved the pellet directly to its mouth) on the third step of the right staircase (the step most frequently used). Data were collected the eighth week post-transplantation. In the event that no successful grasping was found, we used the data from the seventh week post-transplantation. One control, two lesioned, two intrastriatal-, and four intranigral-grafted mice were excluded from the analysis because they did not exhibit any successful grasping on the third step of the staircase. The time required to complete each task phase was calculated according to the following formula: (number of frames constituting the phase − 1) × 40 ms. Analysis of the length of the reaching movement trajectory was performed after importing corresponding frames into ImageJ 1.48V software (NIH, Bethesda, MD, USA). The pellet position on the third step of the staircase was used as a reference point then the tip of the third finger was pointed on the successive frame, and the x/y coordinates were determined, allowing us to draw Cartesian trajectories of each reaching movement [61,62].

### 2.4. L-DOPA Administration and Behavioral Assessment

In order to test the effects of a single injection of L-DOPA on motor-skilled reaching and grasping abilities, the staircase test was performed on two consecutive days (16 weeks post-grafting). L-DOPA methyl ester (10 mg/kg, Sigma-Aldrich, Saint Louis, MO, USA) and the peripheral DOPA decarboxylase inhibitor benserazide-HCl (10 mg/kg, Sigma-Aldrich) were freshly dissolved in 0.9% saline immediately prior to use. On the first day, the mice were tested on the staircase for 10 min in order to measure motor performances 16 weeks after transplantation. On the second day, the mice were injected with drugs (0.1 mL/10 g body weight, i.p.) 30 min prior to placing them on the staircase for a 10 min session in order to cover the period of maximal drug effect. During the analysis of the skilled reaching abilities, one lesioned and two intranigral-grafted mice were excluded from the count because they did not demonstrate entry into the corridor of the device.

### 2.5. Electrophysiological Measurements

Before surgery, the mice were deeply anesthetized with urethane (1.8 g/kg) injected i.p. before being secured to a stereotaxic frame and maintained at 37–38 °C with a heating pad. A mouse brain stereotaxic atlas [63] was used to guide electrode and pipette placements. Throughout the experiment, the efficiency of the anesthesia was determined by examining the tail-pinch reflex. Additional urethane (0.25 g/kg, i.p.) was administered when necessary.

Electrophysiological single-unit activity was recorded in the dorsolateral striatum (DLS) using electrodes pulled from borosilicate glass capillaries (GC 150 F, Harvard Apparatus, Holliston, MA, USA) with a P-97 Flaming Brown micropipette puller (Sutter Instrument, Novato, CA, USA). The tip of the electrode was broken to a diameter of 2 µm, and the electrode was filled with 0.4 M NaCl solution containing 2.5% neurobiotin (Vector Laboratories, Burlingame, CA, USA). Electrodes had an in vivo resistance of 12–20 MΩ. Recording electrodes were lowered in the DLS ipsilateral to the lesion or transplantation site (2 to 2.2 mm lateral and 0.9 to 0.6 mm anterior to bregma) at a depth comprised between 2 mm and 3 mm from the brain surface.

Neuronal activity was amplified 10 times, filtered (bandwith: 300 Hz–10 kHz), and further amplified 100 times (MultiClamp 700B, Axon Instruments, San Jose, CA, USA). The signal was digitized (Micro1401 mk II, Cambridge Electronic Design Ltd., Cambridge, UK) and acquired on a computer using Spike2 software. Recorded neurons were juxtacellularly labeled with neurobiotin (Vector Laboratories), as described elsewhere [64]. Briefly, positive 250 ms current pulses were applied at 2 Hz with increasing currents (1 to 5 nA) until driving cell firing for at least 5 min. Immediately after neurobiotin injection, the mice were transcardially perfused with 0.9% NaCl followed by cold 4% paraformaldehyde (PFA in 0.1 M PB; pH 7.4). The brains were collected and post-fixed for 24 h at 4 °C in 4% PFA and cryoprotected overnight in sucrose (30% in 0.1 M PB) at 4 °C. Serial coronal sections (40 µm) were cut using a cryostat (CM3050 S, Leica Biosystems, Nussloch, Germany). To reveal neurobiotin, the sections were rinsed three times in 0.1 M phosphate buffer saline (PBS), processed for 1 h with a blocking solution (3% BSA and 0.3% Triton X-100 in PBS), and incubated overnight at 4 °C in Streptavidin Alexa Fluor 568 (Life Technologies, Carlsbad, CA, USA) diluted at 1:800 in PBS containing 3% BSA and 0.3% Triton X-100. The sections were then rinsed three times in PBS before being mounted on gelatin-coated slides, air-dried, and coverslipped with DePeX (Sigma-Aldrich).

Orthodromic stimulations of the cortex ipsilateral to the recording site were performed using a concentric bipolar electrode (SNEX-100, Rhodes Medical Instruments Inc., Summerland, CA, USA) implanted in the motor cortex in order to evaluate the response of striatal neurons to cortical input. Stimulations (duration: 0.5 ms) were applied every 3 s using an external stimulator (DS3, Digitimer, Letchworth Garden City, UK) triggered by a 1401 Plus system (Cambridge Electronic Design Ltd.), as previously reported [65,66,67].

The recordings were analyzed offline using Spike2 7.0 (Cambridge Electronic Design Ltd.). The action potential duration was measured from the positive to the negative pic. In order to assess neuronal activity, the action potential mean firing rate was measured for 1 min periods. The stimulation threshold was determined by measuring the minimal stimulation intensity required to evoke a spike response with a probability close to 50%.

### 2.6. Tissue Processing and Immunohistochemistry

Tissue processing: The mice were anesthetized and transcardially perfused with 150 mL saline (0.9%) followed by 200 mL ice-cold PFA (4%). The brains were removed and postfixed overnight in 4% PFA and then cryoprotected overnight in sucrose (30% in 0.1 M PB). The brains were cut into 6 series at a thickness of 40 μm on a freezing microtome (Microm HM450, Thermo Fisher Scientific, Waltham, MA, USA) and stored in a cryoprotective solution (20% glucose, 40% ethylene glycol, 0.025% sodium azide, and 0.05 M PB; pH 7.4).

Immunohistochemistry: Free-floating sections were washed in PBS (0.1 M; pH 7.4) and incubated for 1 h at room temperature (RT) in a blocking solution containing 5% BSA and 0.3% Triton X-100 in PBS 0.1 M at pH 7.6. Primary antibodies diluted in blocking solution were applied for 2 h at RT and overnight at 4 °C. After three washings in PBS, appropriate secondary antibodies coupled to Alexa Fluor fluorochromes (Life Technologies, Carlsbad, CA, USA) were diluted in PBS and applied for 1 h at RT. The sections were rinsed and mounted on gelatinized slides before being coverslipped with DePeX (Sigma-Aldrich) mounting medium. The following antibodies were used in this study: goat anti-GFP (1/1000; Abcam, Cambridge, UK), rabbit anti-GFP (1:1000; Life Technologies), chicken anti-GFP (1:1000, Abcam), mouse anti-TH (1:5000; ImmunoStar, Hudson, WI, USA), rabbit anti-Girk2 (1:100; Sigma-Aldrich), rabbit anti-calbindin (1:5000; Swant, Burgdorf, Switzerland), rabbit anti-DAT (1:5000; a generous gift from B. Bloch, CNRS UMR 5227 Bordeaux-France), mouse anti-NeuN (1:500; Sigma-Aldrich), chicken anti-GFAP (1:1000; Abcam), and rabbit anti-Iba1 (1:500; Fujifilm Wako, Osaka, Japan) Rabbit anti-CD86 (1:200; Abcam) and goat anti-Arg1 (1:250; Santa Cruz Biotechnology, Dallas, TX, USA.) were used for neurotoxic and neuroprotective phenotypes, respectively. Rat anti-C3 (1:200; Abcam) and rabbit anti-CD109 (1:200; Abcam) were used for A1 and A2 phenotypes, respectively.

Electron microscopy: The mice were perfused transcardially with 150 mL of saline (0.9%) followed by 200 mL fixative (2% PFA and 0.2% glutaraldehyde in 0.1 M PB; pH 7.4). The brains were postfixed overnight in 2% PFA and cut into 60 µm coronal sections with a vibrating microtome (Microm HM650V, Thermo Fisher Scientific). The sections were preincubated for 90 min in a blocking solution (0.2% acetylated BSA (BSAc and Aurion, Wageningen, The Netherlands) in 0.1 M PBS; pH 7.4). For immunogold detection of GFP, the vibratome sections were incubated overnight at 4 °C with rabbit anti-GFP (1:1000; Life Technologies), rinsed in PBS, and incubated for 4 h at RT in goat anti-rabbit IgG conjugated to ultra-small gold particles (0.8 nm, 1:100; Aurion). The immunogold signal was intensified using a silver enhancement kit (HQ Silver; Nanoprobes, Yaphank, NY, USA). After several washes, the sections were postfixed in 2% osmium tetroxide and dehydrated in an ascending series of dilutions of ethanol, including 70% ethanol containing 1% uranyl acetate and propylene oxide. Then, the sections were impregnated in LR White resin overnight (Sigma-Aldrich), mounted on glass slides, and cured at 60 °C for 24 h. Areas of interest were first visualized in a light microscope and then cut out from the slide and glued to blank cylinders of resin. Serial ultrathin (60 nm) immunostained sections were cut with a UC6 Ultramicrotome (Leica Biosystems), contrasted with lead citrate, and observed with an electron microscope (JEOL Ltd., Tokyo, Japan)

### 2.7. Data Acquisition and Quantification

Immunofluorescence-treated sections were examined with an Axio Imager M2 Apotome microscope (Carl Zeiss, Oberkochen, Germany). The selected sections were further analyzed and photographed with an FV1000 confocal laser-scanning microscope (Olympus, Tokyo, Japan). The extent of dopaminergic depletion was analyzed by quantifying the number of TH+ neurons remaining in injured SNpcs compared to intact SNpcs in a 1:6 series of sections (*n* = 9). To estimate the total number of dopaminergic neurons found in the transplants, an extrapolation of the total number of GFP+ cells contained in one series of the sections from intrastriatal (*n* = 10) or intranigral (*n* = 10) grafts immunohistochemically labeled for GFP was sampled using confocal microscopy. The method of Abercrombie (1946) [68] was used to correct for the possibility of double-counting due to cells spanning more than one tissue section. To assess the number of dopamine neurons that co-expressed Girk2 within the transplant, one series of the sections from intrastriatal (*n* = 7) or intranigral (*n* = 6) grafts immunohistochemically labeled for GFP and Girk2 was sampled using confocal microscopy, and co-localization was confirmed by z-axis analysis. The same procedure was used to assess the number of GFP+ cells that co-expressed calbindin or dopamine transporter (DAT) in grafts immunohistochemically labeled for these markers. GFP+ fibers were quantified for each animal in five sections for each area using a high-magnification objective (20×).

### 2.8. Statistical Analysis

All data were expressed as mean ± the standard error of the mean (SEM). Normality was checked using the Shapiro–Wilk test. The Mann–Whitney U test was used to compare the average number of GFP+ and NeuN+ cells in the two groups of grafted animals (intranigral- vs. intrastriatal-grafted mice). For the statistical significance of glial fibrillary acidic protein (GFAP)+ and ionized calcium binding adaptor molecule 1 (Iba1) + cells in the host, transplant, and two grafted groups, the Holm–Sidak method was used with alpha = 0.05. For behavioral studies, the following statistical tests were performed: (i) a Kruskal–Wallis test to compare the time required to perform movements and the staircase test with or without L-DOPA injections; (ii) a one-way analysis of variance (ANOVA) with group as a factor for the challenging beam test, the staircase test, and analysis of the length of the reaching-movement trajectory 8 weeks post-transplantation; and (iii) repeated measures of ANOVA using group (for the cylinder test and the analysis of the length of the reaching-movement trajectory in the L-DOPA condition) or time (rotation tests) as factors. Post hoc analysis was conducted with Dunn’s model for nonparametric tests and Fisher’s protected least significance difference (PLSD) test or the Dunnett test, as appropriate.

For electrophysiological experiments, statistical analyses were performed using a one-way ANOVA followed by a Tukey post-hoc test. The statistical significance level was set at *p* < 0.05.

## 3. Results

### 3.1. Both Intranigral and Intrastriatal Transplants Express Dopaminergic Markers

Four months after 6-OHDA injection (*n* = 9), the number of TH+ neurons in lesioned SNpcs was 114 ± 41 compared to 1259 ± 29 in the intact side (91.4 ± 3% decrease), which is in accordance with our previous findings [37]. Grafts survived in 97% of the intranigral (*n* = 21) and intrastriatal (*n* = 22) transplants. Intranigral grafts were mainly located in the dorsal part of the SNpc, and in some cases, a few GFP+ cells were also found along the injection tract. Intrastriatal grafts were located in the dorsal or medial parts of the striatum near the lateral ventricle. Some isolated GFP+ cells were observed at the host–graft interface. Intrastriatal grafts (*n* = 10) contained significantly less (−48.72%) GFP+ neurons (732 ± 231) than intranigral ones (1426 ± 208, *n* = 10) (*p* < 0.05; Mann–Whitney test). Almost all GFP+ cells within the intranigral or intrastriatal transplants co-expressed TH (>95%), which is in agreement with previous studies [39,69].

### 3.2. Dopamine Neurons within the Grafts Express SNpc and VTA Markers

While the vast majority of VTA dopamine neurons expressed the calcium-binding protein calbindin [70,71,72,73], dopamine neurons of the SNpc almost exclusively expressed the Girk2 protein [13,69,74,75]. It has been shown in embryonic mice [69] and in humans [13] that dopamine neurons in VM grafts transplanted in host striatum that co-expressed Girk2 innervated the host striatum, while dopamine neurons derived from VTA that co-expressed calbindin did not innervate the striatum to a great extent.

Here, we aimed at determining the proportion of dopamine neurons for the SNpc vs. VTA subtypes within the grafts. For this, we examined the expression of Girk2 or calbindin within the intrastriatal (*n* = 7) and intranigral (*n* = 6) grafts (Figure 1, Table 2). In both grafts, the GFP+ neurons co-expressing Girk2 were preferentially found on the periphery of the transplant (Figure 1c,h); these neurons had an angular morphology and were larger than GFP+/calbindin+ neurons (Figure 1d,e), a feature characteristic of SNpc dopamine neurons. GFP+ neurons co-expressing calbindin had a more round morphology (Figure 1i,j), were comparatively smaller, round in shape, located mainly within the center of the graft (Figure 1b,g), and more characteristic of VTA dopamine neurons.

More than half of the GFP+ dopaminergic neurons were positive for Girk2 in both transplantation conditions. Indeed, with the intrastriatal transplants, we found that among 653 ± 156 GFP+ neurons, 345 ± 85 neurons were also positive for Girk2, and among 342 ± 74 of GFP+ neurons, 154 ± 37 neurons were positive for calbindin (Figure 1k,l). With the intranigral transplants, among 1261 ± 197 GFP+ neurons, 690 ± 116 neurons were positive for Girk2, and among 694 ± 123 GFP+ neurons, 335 ± 65 were positive for calbindin (Figure 1k,l). However, the mean percentage of Girk2+ neurons was similar between the intrastriatal and intranigral grafts (52.2 ± 0.8% and 54.2 ± 0.9%, respectively), as was the mean percentage of calbindin neurons (44.1 ± 2.29% and 47.4 ± 1.9%, respectively) (Figure 1m,n). Hence, the proportion of GFP+ neurons expressing either calbindin or Girk2 was not significantly different (*p* = 0.346 and *p* = 0.137, respectively; Mann-Whitney test) in both transplantation conditions.

The lower number of TH-GFP+ cells in intrastriatal grafts led us to delve further into the cellular composition of the grafts. In order to identify all transplanted cells after grafting, VM donor tissues were isolated from transgenic mice over-expressing GFP under the influence of beta-actin promoter.

In the first set of experiments, we investigated the proportion of the GFP+ cells that were differentiated into mature neurons by combining NeuN and GFP immunostaining in intranigral (*n* = 6) and intrastriatal (*n* = 6) grafts. We reported a significant increase in the number of NeuN+ cells in the intranigral grafts (1490 ± 194) compared to the intrastriatal grafts (331 ± 66, ** *p* < 0.002) (Figure 2a–g, Table 2).

In the second set of experiments, we used GFAP to detect astrocytes in the host tissue surrounding the grafts and within the grafts. We observed a significant increase in the number of GFAP+ cells in the host tissue surrounding the intrastriatal grafts (host intrastriatal: 471 ± 80) compared to the intranigral grafts (host intranigral: 183 ± 52, *n* = 6, * *p* < 0.01). We also observed a significant increase in the number of GFAP+ cells in the intrastriatal grafts (622 ± 124) compared to the intranigral grafts (207 ± 30, *n* = 6, ** *p* < 0.008) (Figure 2h–m). In order to determine the origin of the GFAP+ cells within the graft, we combined GFAP and GFP labelling. We found a higher number of GFAP+/GFP+ cells in intrastriatal grafts (137 ± 38) compared to intranigral grafts (33 ± 8, *n* = 6, ** *p* < 0.006) (Figure 2n), indicating that more grafted cells were differentiated into astrocytes in intrastriatal grafts.

As the number of astrocytes was different in relation to graft placement, we investigated whether the level of inflammation in the host tissue surrounding the graft and within the graft was different between the two transplanted conditions. For this, we used an Iba1 antibody to detect microglia cells and observed a significant increase in the number of Iba1+ cells in the host tissue surrounding the intrastriatal grafts (317 ± 47) compared to the intranigral grafts (197 ± 19, *n* = 6, * *p* < 0.05) (Figure 2o). There was also a significant increase in the number of Iba1+ cells in the intrastriatal grafts (323 ± 44) compared to the intranigral grafts (158 ± 18, *n* = 6, ** *p* < 0.006) (Figure 2o). We further performed double labeling of Iba1 with GFP to identify the origin of Iba1+ cells within grafts. We found that, for both graft locations, nearly all the Iba1+ cells in the graft were negative for GFP, suggesting infiltration of the host microglia into the graft.

### 3.3. Astrocyte and Microglia Polarization in the Grafts

As the level of inflammation within the graft was different between the intrastriatal and intranigral transplants, we investigated the pro-inflammatory vs. anti-inflammatory profiles of activated microglia and astrocytes in the grafts. Microglial cells respond to brain injury by becoming activated and shifting between neurotoxic (detected with CD86) or neuroprotective phenotypes (detected with arginase-1). We observed that the percentage of Iba1+ cells co-expressing CD86 was higher in intrastriatal grafts (70 ± 36%, Figure 3a–d,a’–d’) compared to intranigral grafts (6 ± 2.8%, Figure 3e–h,e’–h’, *p* < 0.006). We also observed a higher number of Iba1+ cells co-expressing Arg1 in intranigral grafts (20.4 ± 9.7%, Figure 3m–p,m’–p’) compared to intrastriatal grafts (3.7 ± 1.9%, Figure 3i–l,i’–l’, Table 2, *p* < 0.05).

Reactive astrocytes can be divided into toxic A1 astrocytes (detected with Complement 3 (C3)), which induce the rapid death of neurons and oligodendrocytes, and neuroprotective A2 astrocytes (detected with CD109), which promote neuronal survival and tissue repair [76,77]. We observed a significant increase in the percentage of GFAP+ cells co-expressing the C3 marker in host tissue surrounding the intrastriatal grafts (41 ± 8%, Figure 4a–d) compared to the intranigral grafts (23 ± 7%, Figure 4e–h, * *p* < 0.05), and the percentage of GFAP+ cells co-expressing the C3 marker was higher in intrastriatal grafts (44 ± 6%, Figure 4a–d,a’–d’) compared to intranigral grafts (27 ± 4%, Figure 4e–h,e’–h’, * *p* < 0.05). The percentage of GFAP+ cells co-expressing the CD109 marker was very low and not significantly different between the two graft conditions (intrastriatal graft: 6 ± 2%, Figure 4i,k,l; intranigral: 10 ± 6%, Figure 4m,o,p, Table 2).

### 3.4. Projection of Grafted Neurons

GFP fibers from the intrastriatal (Figure 5a–c) or intranigral (Figure 5d–f) grafts were dopaminergic, as the vast majority also co-expressed DAT. We observed that both types of grafts sent projections to appropriate dopaminergic targets (Table 2). However, the pathways used by these axons and their densities were different. In the striatal transplantation position, GFP fibers were found leaving the graft and running through the grey matter (data not shown) of the host striatum. Dense GFP+ fibers were found covering the entire rostrocaudal extent of the striatum (Figure 5 and Figure 6j–m). GFP+ fibers were also present in the nucleus accumbens (Figure 6j,k), frontal cortex (Figure 6h,i), and perirhinal cortex (Figure 6i–l). In the nigral transplantation position, grafted neurons sent projections along trajectories resembling those of the intrinsic dopaminergic nigrostriatal and mesolimbocortical pathways. Indeed, GFP+ fibers extended rostrally from the graft core through the median forebrain bundle (Figure 6f) and the nigrostriatal pathway, globus pallidus (Figure 6f), exiting the dorsal striatum, which was the main target of nigral dopamine neurons (Figure 5d–f and Figure 6b–e). GFP fibers were also detected in VTA target areas, including the nucleus accumbens (Figure 6c) and the septum (Figure 6c,d), as well as the medial prefrontal cortex (Figure 6a–c). We performed a quantitative analysis of the axonal projections from VM grafts placed in the SNpc or striatum. In the nigral graft position, a large number of GFP fibers were identified within the mfb (624 ± 142). Both intranigral and intrastriatal grafts sent projections to the striatum (intrastriatal: 3093 ± 892; intranigral: 1255 ± 352), nucleus accumbens (intrastriatal: 2053 ± 764; intranigral: 831 ± 183), and frontal cortex (intrastriatal: 354 ± 91; intranigral: 129 ± 43). In both grafted groups, GFP+ fibers were still present in the target areas up to sixteen weeks post-grafting.

### 3.5. Establishment of Reciprocal Synaptic Contacts between Intrastriatal or Intranigral Transplants and Host Neurons

In order to investigate to which level transplanted dopamine neurons integrated into the host circuits, we assessed the reciprocal synaptic contacts between intrastriatal or intranigral transplants and the host by electron microscopy four months post-grafting (Figure 7, Table 2). Host neurons developed axosomatic contacts on GFP+-transplanted neurons both in intrastriatal (Figure 7a) and intranigral (Figure 7c) grafts. In addition, we found that transplanted neurons developed symmetric synaptic contacts mainly on dendritic spines and shafts of the striatal neurons in both graft locations (Figure 7b,d). We also observed the presence of axosomatic contacts only at the intrastriatal graft location.

### 3.6. Functional Recovery of Lesioned Mice with Either Intranigral or Intrastriatal Transplants

The net contralateral rotation response to apomorphine during a 60 min session was recorded pre-grafting and thirteen weeks after transplantation in the same animals, as shown in Figure 8a. There was a significant effect of time between the groups (week, F_1,28_ = 5.82; *p* < 0.05). Lesioned animals (*n* = 9) did not show any spontaneous improvement in their rotation scores between the first testing session (61 ± 15) and the second one (68 ± 18, *p* = 0.68, NS). On the other hand, the intranigral-transplanted mice (*n* = 11) net contralateral turns were significantly reduced by 55% (20 ± 12) in comparison to their pre-grafting scores (44 ± 18, *p* < 0.05). In the same way, the intrastriatal-transplanted mice (*n* = 11) rotations were also significantly reduced by a similar proportion of 68% than before grafting (18 ± 6 versus 56 ± 24, respectively; *p* < 0.05). In another set of mice, we evaluated amphetamine-induced rotations both before and after transplantation (Figure 8b) during a 60 min session. There was a significant effect of time between the groups (Week, F_1,12_ = 49.73; *p* < 0.0001). Lesioned animals (*n* = 5) did not show any significant difference in their rotation scores between the first testing session (346 ± 49) and the second one (237 ± 56, *p* = 0.074, NS). On the other hand, the intranigral-transplanted mice (*n* = 5) turns were significantly reduced (24.8 ± 14) in comparison to their pre-grafting scores (317 ± 25, *p* < 0.001). In the same way, the intrastriatal-transplanted mice (*n* = 5) rotations were also significantly reduced in comparison to their pre-grafting scores (35 ± 14 vs. 313 ± 46, respectively; *p* < 0.001).

At three and ten weeks post-grafting, the mice were tested for forelimb asymmetries using the cylinder test. We found a difference between the groups (control, *n* = 8; lesioned, *n* = 7; intrastriatal-grafted, *n* = 11; intranigral-grafted, *n* = 10) in the use of the forepaws contralateral to the lesion site (group, F_3,32_ = 4.09, *p* < 0.05) and a significant interaction between the groups and times of testing (group x week, F_6,64_ = 2.66, *p* < 0.05) (Figure 8c). At three weeks post-transplantation, both the lesioned and transplanted mice showed significant deficits in the use of the forepaws contralateral to the lesion site in comparison to control (control vs. lesioned, *p* < 0.001; control vs. intrastriatal and control vs. intranigral, both *p* < 0.05). Ten weeks after transplantation, both lesioned and intrastriatal-grafted mice showed a marked impairment of the contralateral forelimb contacts in comparison to control mice (both, *p* < 0.01). However, no difference was observed between intranigral-grafted and control mice, indicating a recovery in fine motor skills was required for this motor task.

Six weeks after transplantation, we observed a significant group difference in the challenging beam test in the time required to cross the beam (group, F_3,35_ = 3.45, *p* < 0.05). Indeed, 6-OHDA-lesioned mice needed a much longer time to cross the beam than intranigral-grafted (more than 1.4-fold; *p* < 0.05) and control (1.6-fold; *p* < 0.01) mice. No significant difference was found between the control group and either the intrastriatal (*p* = 0.1024) or intranigral (*p* = 0.4079) group (Figure 8d, Table 2). In order to determine forelimb gait and fine motor coordination, we analyzed the percentage of foot slips performed during the cross. Whereas no difference was found between groups in the percentage of foot slips made with forelimbs that were ipsilateral to the lesion site (Figure 8e; group, F_3,35_ = 0.218, *p* = 0.883, NS), one-way ANOVA revealed a main effect between groups in the percentage of foot slips made with forelimbs that were contralateral to the lesion. (Figure 8f; group, F_3,35_ = 4.429, *p* < 0.01). Post-hoc analysis indicated that lesioned mice exhibited a significantly greater percentage of contralateral forelimb errors in comparison to the control group (*p* < 0.05). Intrastriatal-transplanted mice displayed significantly more contralateral forelimb errors than the control group (*p* < 0.01) and were not different from lesioned mice (*p* = 0.7837; NS). Intranigral-grafted mice displayed a significant decrease in contralateral forelimb errors in comparison both to the lesioned and intrastriatal-transplanted groups (both *p* < 0.05) and were not different from control mice (*p* = 0.4931; NS), indicating a restoration of contralateral forelimb use following intranigral transplantation.

Lesions of the nigro–striatal pathway produced skilled-forelimb reaching deficits that we assessed here in four groups of animals (control, *n* = 8; lesioned, *n* = 9; intrastriatal-grafted, *n* = 11; intranigral-grafted, *n* = 11). Prior to surgery, the animals were trained on the staircase apparatus in order to assess baseline performance and were thereafter tested once a week during six consecutive sessions (from weeks three to eight post-transplantation) divided into two blocks of three sessions (from weeks three to five and from weeks six to eight). No significant difference between groups was found in the baseline performance (all *p* > 0.05, NS), indicating that all groups acquired the test similarly (Figure 9a, left panel). Starting at three weeks following the transplantation surgery and until the fifth week post-transplantation, all lesioned (29.01 ± 4.97), intranigral- (35.20 ± 6.84), and intrastriatal- (37.68 ± 7.26) grafted mice exhibited severe and significant impairment (F_3,35_ = 2.899, *p* < 0.05, Figure 9a, middle panel) of contralateral skilled-limb movements compared to the control group (57.30 ± 6.53, all *p* < 0.05). Six to eight weeks post-transplantation, one-way ANOVA revealed difference between the groups (F_3,35_ = 4.211, *p* < 0.05, Figure 9a, right panel), indicating that forelimb reaching deficits were still present for both lesioned and intrastriatal-grafted mice (both, *p* < 0.01) compared to the control group. However, intranigral-grafted mice showed a significant improvement in grasping performance compared to lesioned and intrastriatal-transplanted mice (both, *p* < 0.05), and no difference was found between control and intranigral-grafted mice (*p* = 0.47).

We evaluated the kinetics of forelimb movements contralateral to the lesion site from seven to eight weeks post-transplantation. For this, the complete movement was divided into three tasks (reaching, grasping, retrieval), and we measured the time required to complete each task phase on the third step of the staircase (the step commonly reached by the majority of animals). We found that the time required to achieve the reaching phase was different between the groups (Figure 9b, *p* < 0.001; Kruskal–Wallis test). In particular, lesioned mice (*n* = 7) needed more time than both control (*n* = 7) and intranigral-grafted (*n* = 7) mice (*p* < 0.01 and *p* < 0.05, respectively) to contact the pellet. In addition, this reaching phase was also significantly longer with the intrastriatal-grafted mice (*n* = 9) in comparison to both control and intranigral-grafted mice (both, *p* < 0.05). In contrast to the reaching time, the time taken to execute the grasping, the retrieval, or the complete movement was similar between the groups (Figure 9c–e; *p* = 0.752, *p* = 0.302, and *p* = 0.890, respectively). In conclusion, we found that the time required to achieve the reaching phase was similar between control and intranigral-grafted mice, indicating a marked recovery of movement in this task.

In order to refine this initial reaching phase, we analyzed the reaching trajectories of each group (Figure 9f). Interestingly, while the reaching trajectories of both control and intranigral-grafted mice were quite rectilinear, both lesioned and intrastriatal-grafted mice demonstrated very irregular reaching trajectories. To quantify these trajectories, we computed the length of the reaching-movement trajectories expressed in mm and found a significant difference between groups (Figure 9g; F_3,26_ = 10.39, *p* < 0.001; one-way ANOVA), with the lesioned mice displaying a significant increase in the length of the trajectories compared to both control and intranigral-transplanted mice (*p* < 0.001 and *p* < 0.01, respectively). Following the same pattern, the trajectories of intrastriatal-grafted mice were also significantly longer than in both control and intranigral-transplanted mice (both, *p* < 0.001). Interestingly, no difference was found in the length of trajectories between both control and intranigral (*p* = 0.52) mice or between lesioned and intrastriatal-grafted (*p* = 0.69) mice. This indicated that only intranigral grafts were able to restore this motor behavior.

We then analyzed the length of reaching-phase trajectories following a single i.p. L-DOPA injection (Table 2). For this purpose, we performed the staircase test for one session per day on two consecutive days (without and with an L-DOPA injection) 16 weeks post-grafting in all groups of the animals (control, *n* = 8; lesioned, *n* = 8; intrastriatal-transplanted, *n* = 11 and intranigral-transplanted, *n* = 9, Figure 10a). Sixteen weeks after transplantation, the first session revealed a difference within groups (Figure 10a, left panel; *p* < 0.05, Kruskal–Wallis test). We found a robust 6-OHDA lesion effect in the contralateral forelimb bias between lesioned and control mice (*p* < 0.05). Additionally, only intranigral-grafted mice displayed significant improvement of their contralateral forelimb bias in comparison to lesioned mice (*p* < 0.01), and no significant difference was found between control and intranigral-transplanted mice (*p* = 0.4947). Interestingly, a single injection of L-DOPA improved the performance of the reaching task with impaired limbs of intrastriatal-transplanted mice. The contralateral bias of lesioned and intrastriatal-grafted mice increased from 17.5 ± 6.66% to 34.37 ± 8.09% and from 34.54 ± 12.89% to 43.18 ± 9.59%, respectively, following the L-DOPA injection (Figure 10a, right panel). Despite a marked impairment in the contralateral forelimb of lesioned mice compared to control mice (*p* < 0.05), no difference was found among all the other groups.

We next conducted a detailed analysis of the grasping movement for each group during the two sessions (without and with an L-DOPA injection) in the staircase test. Sixteen weeks post-grafting and without an L-DOPA injection, we found a similar pattern in the grasping trajectories as described above with very irregular reaching movements for both lesioned (*n* = 7) and intrastriatal-grafted (*n* = 7) mice in comparison to control (*n* = 7) and intranigral-grafted (*n* = 7) mice (Figure 10b). Conversely, after a single L-DOPA injection, no modification was observed in control, lesioned, or intranigral-grafted mice, whereas intrastriatal-grafted mice showed more regular grasping movements (Figure 10c). Repeated two-way ANOVA measures on the length of the trajectories revealed a main effect between groups (Figure 10d; group, F_3,24_ = 10.98, *p* < 0.0001). Without an L-DOPA injection, both lesioned and intrastriatal-grafted mice displayed a significantly longer length of trajectory in comparison to control (*p* < 0.001 and *p* < 0.01, respectively) and intranigral-grafted mice (*p* < 0.001 and *p* < 0.05, respectively). However, following a single L-DOPA injection, no difference was found between intrastriatal-grafted mice and control or intranigral-grafted mice (*p* = 0.532 and *p* = 0.317, respectively). In contrast, the length of trajectory for the lesioned mice was not affected by the treatment and was significantly longer than control, intrastriatal-, and intranigral-transplanted mice (*p* < 0.05, *p* < 0.05, and *p* < 0.01, respectively). This set of experiments indicated that only intranigral grafts restored normal behavior in this task, and this was not further potentiated by an L-DOPA injection. Intrastriatal grafts were deficient but were able to demonstrate this behavior restored to normal levels by a single L-DOPA treatment.

### 3.7. Removal of Intranigral Graft Abolishes Functional Recovery

In order to confirm that the functional recovery was induced by intranigral grafts, we proceeded to remove the intranigral-grafted transplant by an injection of 6-OHDA 2 months after transplantation. The functional impact of the graft lesion was assessed 4 weeks after 6-OHDA injection using amphetamine-induced rotation. In transplanted-then-lesioned animals, the original motor asymmetry deficit re-appeared, and the number of rotations was no longer different from the performance observed in the pre-graft rotation scores (Figure 11d). Histological evaluation of the brain 3 weeks after the injection of 6-OHDA into the graft showed an almost-disappearance of the GFP signal in the transplant (Figure 11a,b), as well as in brain areas normally innervated by the graft, such as the striatum and the cortex. Our results showed that a second lesion induced the degeneration of the grafted dopaminergic neurons and projections and reversed the improvement in rotation behavior induced by the intranigral transplants, confirming that the functional recovery was indeed induced by the intranigral graft.

### 3.8. Electrophysiological Assessment of Striatal Projection Neurons

Dopaminergic denervation resulted in an alteration of striatal neuron basal-firing properties, as well as in cortico-striatal information processing [78,79]. In order to assess the ability of the two grafting locations to restore striatal neuron function, we recorded the electrophysiological activity of striatal projection neurons using single-unit recordings in anesthetized mice. We evaluated the spontaneous firing frequency and the response of striatal neurons to cortical stimulations in control, lesioned, intrastriatal- and intranigral-grafted mice (*n* = 14, *n* = 8, *n* = 8, and *n* = 8 mice, respectively).

Striatal projection neurons represent 90% of total striatal population. They can be electrophysiologically identified by their slow firing rate and their action potential duration, which is intermediate between fast spiking interneurons and cholinergic interneurons (Figure 12a). As the firing rate was expected to be altered by the manipulation of dopaminergic transmission [78,79], we categorized neurons based on their action potential duration. For this purpose, a distribution histogram of action potential duration was performed for all neurons recorded (Figure 12b). Only neurons exhibiting an action potential duration between 0.6 and 1.1 ms were included in this study and considered as putative projection neurons (Figure 12b), and neurons with an action potential duration that were represented in less than 5% of the neuronal population were excluded. Spontaneous firing frequency recorded from striatal neurons increased (one-way ANOVA, *p* = 0.0041) in the lesioned group (0.86 ± 1.47) compared to the control (0.03 ± 0.11) and both grafted groups (intrastriatal: 0.07 ± 0.19; intranigral: 0.02 ± 0.06) (Figure 12c,d). This suggests that the increased excitability observed in striatal neurons following dopamine denervation was restored after both grafting procedures.

In order to assess changes in cortico-striatal transmission, we evaluated the response of striatal neurons to cortical stimulations by measuring the threshold intensity required to evoke a spike response with a probability close to 50%. The cortical stimulation threshold was increased (one-way ANOVA, *p* = 0.0006) in the lesioned and striatal-grafted groups (respectively, 1.26 ± 0.48 and 1.15 ± 0.51) compared to control and SNpc-grafted animals (respectively, 0.63 ± 0.13 and 0.55 ± 0.23) (Figure 12e). These sets of findings indicated that, while both intrastriatal and intranigral transplantation protocols induced the recovery of striatal firing properties, only intranigral transplantation efficiently normalized cortico-striatal responses (Table 2).

## 4. Discussion

The purpose of the present study was to compare the effects of intranigral versus intrastriatal transplantation of fetal VM tissue in a 6-OHDA mouse model of PD. We analyzed graft survival, integration, expression of dopaminergic markers, and projections, as well as behavioral and electrophysiological activity. Our data showed that: (1) fetal VM tissue transplanted either into the SNpc or the striatum is able to survive, re-innervate the denervated striatum, express dopaminergic markers in same proportions, and establish reciprocal synaptic contacts with host neurons. (2) Both graft locations significantly reduced apomorphine- and amphetamine-induced rotations, ameliorated motor performance of simple behavioral tasks, and (3) induced the recovery of striatal firing properties of MSN. In contrast, only intranigral transplantation (4) dramatically increased the number of dopaminergic neurons found within the transplant, (5) sent appropriate projections to the striatum through the mfb, (6) improved recovery of forelimb fine motor skills for complex behavior tasks without further need for L-DOPA treatment, and (7) efficiently normalized cortico-striatal responses.

Homotopic placement of the intranigral transplant may expose the cells to appropriate nigral trophic factors and afferent inputs, allowing the better survival of the grafted neurons, as previously indicated [29,80,81,82]. In line with this hypothesis, two studies on a long-term post-mortem analysis of parkinsonian patients transplanted with fetal nigral neurons in the striatum reported that a very small percentage of fetal nigral neurons developed Lewy body (LB) aggregates. It has been suggested that the presence of LBs in the transplanted cells may be the result of ectopic placement of fetal nigral neurons because, in their ectopic location, the transplanted cells lack normal SN local neural growth-factor support [29,30]. Dopaminergic neurons transplanted either into the striatum or the SNpc were both able to send projections to the striatum. However, only dopaminergic neurons transplanted into the SNpc extended their process through the MFB to the striatum and were able to restore the damaged nigrostriatal pathway in adult mice and, thus, restore physiological circuitry [37,39,41,81].

We observed that the axon terminals of both intranigral- and intrastriatal-grafted dopamine neurons formed synapses on dendritic spines and shafts of striatal neurons. These results are in agreement with findings reported in intact animals [83,84,85]. In addition, we observed the presence of axosomatic contacts between the axons of grafted neurons and host striatal somata only at the intrastriatal graft location.

The distribution pattern of dopaminergic terminals in contact with different parts of the striatal neurons is known to impact synaptic functions. Indeed, most dopaminergic inputs that occur on the necks of dendritic spines seem to be more selective, as they prevent excitatory glutamatergic input to the same spines from reaching the dendritic shaft [83,86]. In contrast, DA terminals that contact the cell somata and proximal dendritic shafts produce a less specific effect mediated by the volumic transmission of dopamine [87].

The anatomical origin of host inputs to grafted neurons is largely influenced by graft placement, as recently shown by Alder et al., 2020 [88]. In their study, they mapped the pattern of host neurons creating synapses with VM cells transplanted in the striatum or in the SN of dopamine-depleted rats. Both intranigral and intrastriatal grafts received appropriate synaptic inputs from the same subtypes of major excitatory (cortical) and inhibitory (striatal and pallidal) populations known to innervate the substantia nigra. Only intrastriatal VM grafts received inappropriate inputs from FOXP2+ arkypallidal neurons, which project normally to the striatum [89,90].

Fetal VM tissue used for transplantation contains both nigral (A9, Girk2+) and VTA (A10, calbindin+) dopamine subtypes. We found that a majority of the dopaminergic neurons grafted both in the striatum and the SNpc expressed the nigral subtype Girk2 in a similar proportion between the two graft conditions. Both in intrastriatal and intranigral transplants, the majority of dopamine neurons that co-expressed Girk2 were found in the periphery of the graft, while dopamine neurons co-expressing the VTA marker calbindin were primarily located within the center of the graft, as previously reported [37,69,91].

Despite the identical number of cells transplanted in the striatum and in the SNpc, we reported a significant increase in the number of NeuN+ cells in the intranigral grafts compared to the intrastriatal grafts. These results suggest a higher rate of cell survival or a lower rate of cell death of neurons in the intranigral position compared to the intrastriatal position and, thus, a better outcome for the homotopic placement of the graft.

Neuroinflammation in PD contributes to disease pathogenesis [92,93,94]. Microglia-mediated dopaminergic neuronal degeneration has been demonstrated in several studies using animal models and was found in the substantia nigra and along the nigrostriatal tract in 6-OHDA rat and mouse PD models, respectively [95,96]. In addition, reactive astrocytes were observed in the substantia nigra and the striatum of a 6-OHDA rat model of PD [97]. We observed a significant increase in the number of microglia cells in the host tissue within and surrounding the intrastriatal grafts compared to the intranigral grafts. Here, we observed for both graft locations that nearly all microglial cells within the graft were negative for GFP, suggesting infiltration of the host microglia into the graft. These results are in agreement with a previous study showing that endogenous microglia of the grafted tissue was rapidly lost after transplantation and that host-derived microglia infiltrated and colonized the graft [98].

Different activation statuses of microglial cells can promote either neurotoxicity or neuroprotection [97,99,100]. In our study, we found that most of the activated microglial cells expressed a pro-inflammatory marker in host tissue surrounding and within the intrastriatal grafts to a greater extent than the intranigral grafts. In addition, we found a higher number of Iba1+ cells co-expressing Arg1 in intranigral grafts compared to intrastriatal grafts. Collectively, these results suggested that the increased number of toxic reactive microglia phenotype and the decreased number of protective reactive microglia phenotype in intrastriatal transplants may promote greater neuronal cell death in intrastriatal grafts compared to intranigral grafts**.**

In addition to microglia, astrocytes also participate in the neuropathology of PD [101,102], where an elevation in the number of astrocytes in the substantia nigra, as well as pathological changes, have been reported postmortem [101,103]. The extent of glial reaction surrounding the graft is negatively correlated with the number of integrated neurons [104]. Reactive astrocytes can be divided into toxic A1 astrocytes, which induce the rapid death of neurons and oligodendrocytes, and neuroprotective A2 astrocytes, which promote neuronal survival and tissue repair [76,77]. We observed that the percentage of GFAP+ cells co-expressing the C3 marker (for A1 astrocytes) was higher in intrastriatal grafts compared to intranigral grafts and that the percentage of GFAP+ cells co-expressing the CD109 marker (for A2 astrocytes) was very low and not significantly different between the two graft conditions. The increase in the number of toxic astrocytes in host tissue surrounding the intrastriatal graft, as well as within the graft, may contribute to the cell death observed in the intrastriatal graft. The processes of glial cell activation and polarization are highly dynamic [100,105], and it may be interesting to perform a time-course study to analyze the evolution of the glial activation and development of grafted neurons to determine whether there is a correlation between the polarization of astrocytes and the survival of grafted neurons.

To evaluate the ability of the two graft strategies to restore physiological striatal activity, electrophysiological investigation of spontaneous striatal activity and cortico-striatal synaptic efficacy were investigated. Our data demonstrated that lesioned animals exhibited increased spike-firing that could be restored to control levels by both striatal and nigral grafts. However, the drop in cortico-striatal transmission efficiency following lesioning was restored only by the striatal graft. Thus, although spontaneous striatal activity was rescued both by striatal and nigral grafts, fine modulation by cortical input was restored only by nigral grafts.

In order to evaluate the integration of the grafted neurons into the host motor circuitries, we investigated the ability of the grafts to restore motor function by using a battery of behavioral tests. For all tests used, lesioning of the nigrostriatal pathway induced strong behavioral deficits. Moreover, both the intrastriatal and intranigral groups showed an improvement of simple motor behavior, such as apomorphine-induced rotation. However, only intranigral-grafted mice displayed significant improvements in fine motor skills evaluated in the staircase test and the trajectories of grasping-movement analysis. Until this study, contrasting results have been previously reported concerning functional recovery following intrastriatal or intranigral VM grafts. For instance, while some investigations have reported partial restoration in drug-induced rotation behavior in rodents after intranigral transplantation [37,106,107,108], only very limited [24,109,110] or no [111,112] improvement on more complex tasks have been reported. Similarly, in the case of intrastriatal transplantations, several studies have reported significant improvements in motor performance for simple tests [19,106,110,112,113,114] but only limited [42,44,45] or no improvement [40,109,115,116,117] in more complex tests, such as the grasping task. In our study, the detailed analysis of the grasping movement demonstrated that the trajectory of the reaching movement observed in the staircase test was more regular for the intranigral-transplanted group compared to the intrastriatal-grafted condition. These results suggest that dopaminergic neurons grafted within the homotopic location allowed better recovery of fine complex movements in contrast to the intrastriatal graft. In line with this observation, a previous study assessing movement components following an intrastriatal graft demonstrated that reaching movements, such as orient, digit close, advance, or pronation, remained impaired [118]. Hence, our findings support the hypothesis that graft location and task complexity are key factors determining the functional efficacy of VM grafts [119].

Intrastriatal-grafted dopamine neurons establish synaptic contacts with the host striatal neurons [120,121], restore dopamine release close to normal level [16,122], and reduce dopamine receptor supersensitivity in the vicinity of the graft [123]. However, despite their ability to re-innervate the host striatum, in their ectopic position, intrastriatal-grafted dopaminergic neurons are unlikely to receive appropriate afferent inputs from the host that are necessary to control fine reaching movement. Indeed, nigrostriatal projections were shown to be necessary to relay specific pattern information involved in the control of complex responses [40,124]. Thus, the incomplete restoration of neural circuitry may be a major contributor to the default in trajectory grasping movement observed in the intrastriatal-grafted group.

The present study demonstrated that skilled forelimb movements were improved after a single L-DOPA injection in intrastriatal-grafted mice but not in lesioned mice. L-DOPA did not produce additional amelioration in intranigral-transplant mice, as this group spontaneously showed similar behavior to the control group. L-DOPA is the most effective and commonly used medication to alleviate motor symptoms for Parkinson’s disease. Previous studies in parkinsonian patients treated with L-DOPA have shown impressive efficacy, reducing tremor [125,126,127,128] or bradykinesia [125,126,128,129]. However, this treatment has failed to significantly improve gait parameters or aspects of reaching movements [130,131,132,133]. The absence of motor behavioral recovery observed in lesioned animals treated with L-DOPA is in line with previous studies showing that skilled reaching movements were more resistant to L-DOPA treatment [134,135] while improving general bradykinesia [136]. One of the assumptions explaining the presence of these remaining deficiencies despite L-DOPA treatment was that restoration of complex motor behavior requires as a precondition the re-establishment of synaptic function. This hypothesis is strengthened by several findings in which, after cell–dopaminergic grafts, behavioral recovery has been evident only in simple, but not in complex, tasks [40,137], suggesting the necessity of combination therapy. Additionally, while intrastriatal transplantation alone does not lead to the restoration of fine motor performance, our results indicated that the combination of intrastriatal transplantation and L-DOPA treatment seems to improve the trajectory of skilled reaching movements. A number of studies have reported that, in 6-OHDA-lesioned rodents, L-DOPA administration has had the capacity to restore the control dopamine level within the substantia nigra, but only partially in the striatum area [138,139,140]. The mechanisms underlying this arise from the significant innervation of the substantia nigra *pars reticulata* (SNpr) from the descending striato-nigral fibers. Specifically, it is now recognized that dendritic dopamine release plays a key role in the regulation of striato-nigral input (via D1-receptors), as well as in nigral output [111,141,142].

In light of these elements, it can be proposed that recovery of fine motor skills requires the near-complete restoration of the nigrostriatal circuitry and that intrastriatal transplantation or L-DOPA treatment alone do not lead to improvements in grasping movements. In contrast, intranigral transplantation may sustain the capacity to ameliorate complex motor behaviors without further need for L-DOPA therapy and its subsequent complications, such as tardive dyskinesia, possibly through dendritic dopaminergic control of SNpr activity and its ability to partially re-innervate not only the denervated striatum but also other nigral targets, such as the subthalamic nucleus or the globus pallidus, as we also reported here. This is of interest, as L-DOPA therapy is notorious for its decreasing beneficial effect with time, a drawback that might not be observed with intranigral transplantation, which restores physiological loss and fine motor skills, as shown here.

Several mechanisms are responsible for the functional improvements observed following neural transplantation. Grafted cells could either directly integrate within the host brain or indirectly affect the host-injured microenvironment. Indeed, several studies, including our own, have demonstrated that grafted neurons integrate neuroanatomically into the host circuitries and promote the reconstruction of damaged connections [36,37,54,81,143,144]. Transplanted cells may also act at the host-injured tissue and may produce neurotrophic factors acting on the remaining or degenerating adult host neurons [145,146]. Grafted cells may also modulate host inflammation and enhance recovery by reducing pro-inflammatory factors [100,147].

In conclusion, by combining analyses of anatomical, electrophysiological, and both qualitative and quantitative aspects of fine motor skills, our results indicated that homotopic placement of immature dopaminergic neurons within the SNpc-lesioned site appeared to be an appropriate strategy to restore physiological functioning of the basal ganglia loops with a subsequent beneficial fine motor behavioral effect and no further need for L-DOPA adjuvant therapy. Whether this homotopic graft placement also restores nonmotor parkinsonian symptoms is yet to be determined.

## Figures and Tables

**Figure 1 cells-11-01191-f001:**
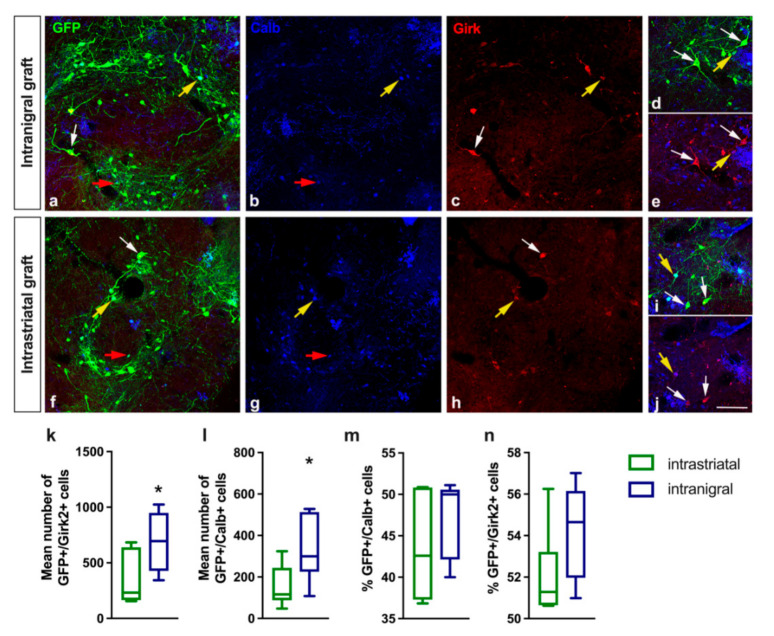
Identification of dopamine neuron subtypes within the intranigral and intrastriatal transplants four months post-transplantation. High magnification confocal photomicrographs of coronal sections showing GFP (GFP; green), calbindin (Calb; blue), and Girk2 (Girk; red) in either intranigral (**a**–**e**) or intrastriatal (**f**–**j**) transplants. The GFP+/calbindin+ double-labelled neurons (red arrows) were located preferentially towards the center of both the intranigral (**a**,**b**) and intrastriatal (**f**,**g**) grafts. These cells were of the typical small and rounded morphology indicative of the A10 dopamine neuron types (**i**,**j**). The GFP+/Girk2+ double-labelled neurons (white arrows) were located preferentially in the periphery of both the intranigral (**a**,**c**) and intrastriatal (**f**,**h**) grafts and were of the typical large and angular morphology (**d**,**e**) indicative of the A9 dopamine neuron type. Yellow arrows point to rare populations of GFP+/calbindin+/Girk2+ cells. Also shown are quantifications of the number of surviving GFP+/Girk2+ (**k**) or GFP+/calbindin+ (**l**) cells within intrastriatal (*n* = 7) and intranigral (*n* = 6) transplants and the percentage of GFP+/calbindin+ (**m**) or GFP+/Girk2+ (**n**) cells within intrastriatal and intranigral transplants. Note that, although there are more surviving GFP+/Girk2+ and GFP+/calbindin+ in intranigral versus intrastriatal transplants, the proportion of these dopaminergic neurons was similar in both transplantation conditions. Data are presented as mean ± SEM. Comparison between the groups was performed by a Mann–Whitney nonparametric test. * *p* < 0.05. Scale bar: 60 µm.

**Figure 2 cells-11-01191-f002:**
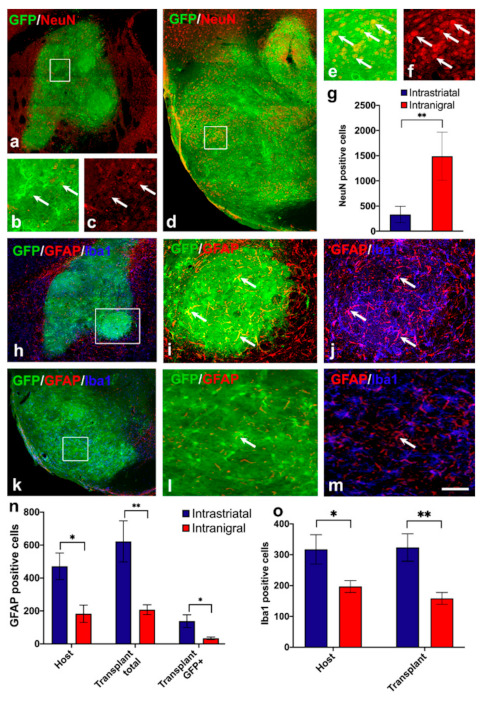
Cellular composition of the grafts. Representative images of NeuN (red) and GFP (green) in intrastriatal (**a**–**c**) and intranigral (**d**–**f**) grafts. (**b**,**c**) are high-magnification views of boxed areas in (**a**); (**e**,**f**) are high-magnification views of boxed areas in (**d**); arrows indicate NeuN+ cells labeled with GFP. Also shown are quantifications of the number of NeuN+ cells in intrastriatal and intranigral grafts and (**g**) representative images of GFAP (red) and GFP (green) in intrastriatal (**h**–**j**) and intranigral (**k**–**m**) grafts. (**i**,**j**) are high-magnification views of the boxed areas in (**h**); (**l**,**m**) are high-magnification views of the boxed areas in (**k**); arrows indicate GFAP+ cells labeled with GFP. Also shown are quantifications of the number of GFAP+ (**n**) or Iba1+ (**o**) cells in the host tissues both surrounding the intrastriatal (*n* = 6) and intranigral (*n* = 6) grafts and within the intrastriatal and intranigral grafts. Data are presented as mean ± SEM. Comparison between the groups was performed by a Mann–Whitney nonparametric test. * *p* < 0.05; ** *p* < 0.01. Scale bar: 235 μm (**a**,**h**,**k**); 65 µm (**b**,**c**,**e**,**f**,**i**,**j**); 40 μm (**l**,**m**).

**Figure 3 cells-11-01191-f003:**
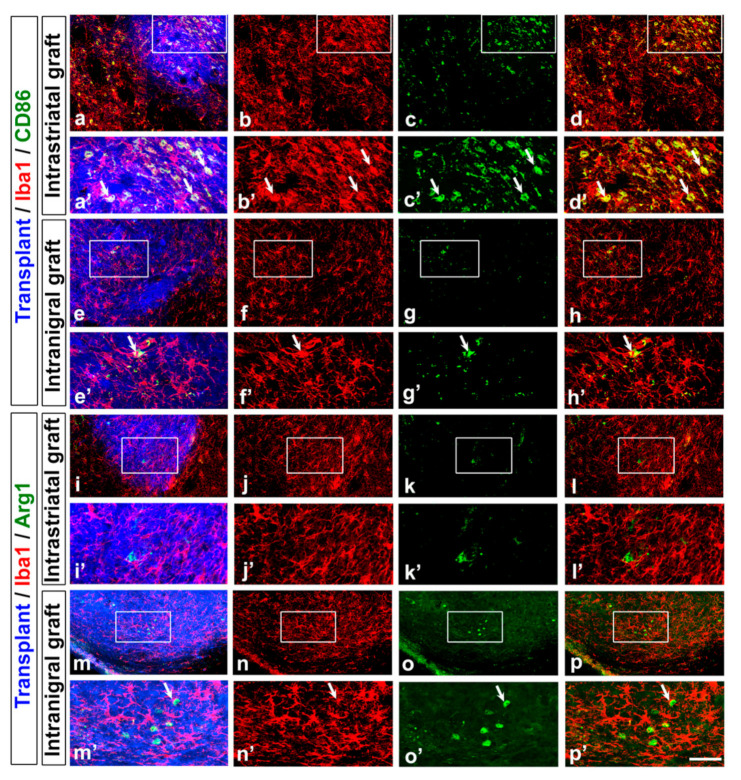
Microglia polarization in the grafts. Representative images of transplant (blue), Iba1 (red), and CD86 (green) in intrastriatal (**a**–**d**) and intranigral (**e**–**h**) grafts. (**a’**–**d’**) are high-magnification views of boxed areas in (**a**–**d**), respectively. Arrows indicate Iba1+ cells within the graft labeled with CD86. Representative images of transplant (blue), Iba1 (red), and Arg1 (green) in intrastriatal (**i**–**l**) and intranigral (**m**–**p**) grafts. (**i’**–**l’**) are high-magnification views of boxed areas, respectively, in (**i**–**l**). (**m’**–**p’**) are high-magnification views of the boxed areas, respectively, in (**m**–**p**). Arrows indicate Iba1+ cells in the graft labeled with Arg1. Scale bar: 360 µm (**a**–**p**) and 115 µm (**a’**–**p’**).

**Figure 4 cells-11-01191-f004:**
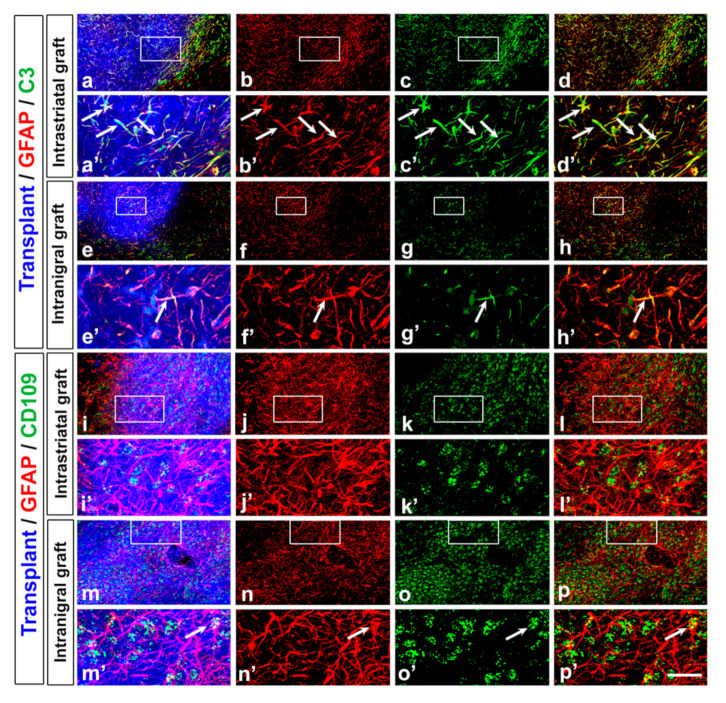
Astrocyte polarization in the grafts. Representative images of transplant (blue), GFAP (red), and C3 (green) in intrastriatal (**a**–**d**) and intranigral (**e**–**h**) grafts. (**a’**–**d’**) are high-magnification views of the boxed areas in (**a**–**d**), respectively; (**e’**–**h’**) are high-magnification views of boxed areas in (**e**–**h**), respectively. Arrows indicate GFAP+ cells in the graft labeled with C3. Representative images of transplant (blue), GFAP (red), and CD109 (green) in intrastriatal (**i**–**l**) and intranigral (**m**–**p**) grafts. (**i’**–**l’**) are high-magnification views of boxed areas in (**i**–**l**), respectively; (**m’**–**p’**) are high-magnification views of boxed areas in (**m**–**p**), respectively. Arrows indicate GFAP+ cells in the graft labeled with CD109. Scale bar: 515 µm (**a**–**p**) and 130 µm (**a’**–**p’**). H: host; T: transplant.

**Figure 5 cells-11-01191-f005:**
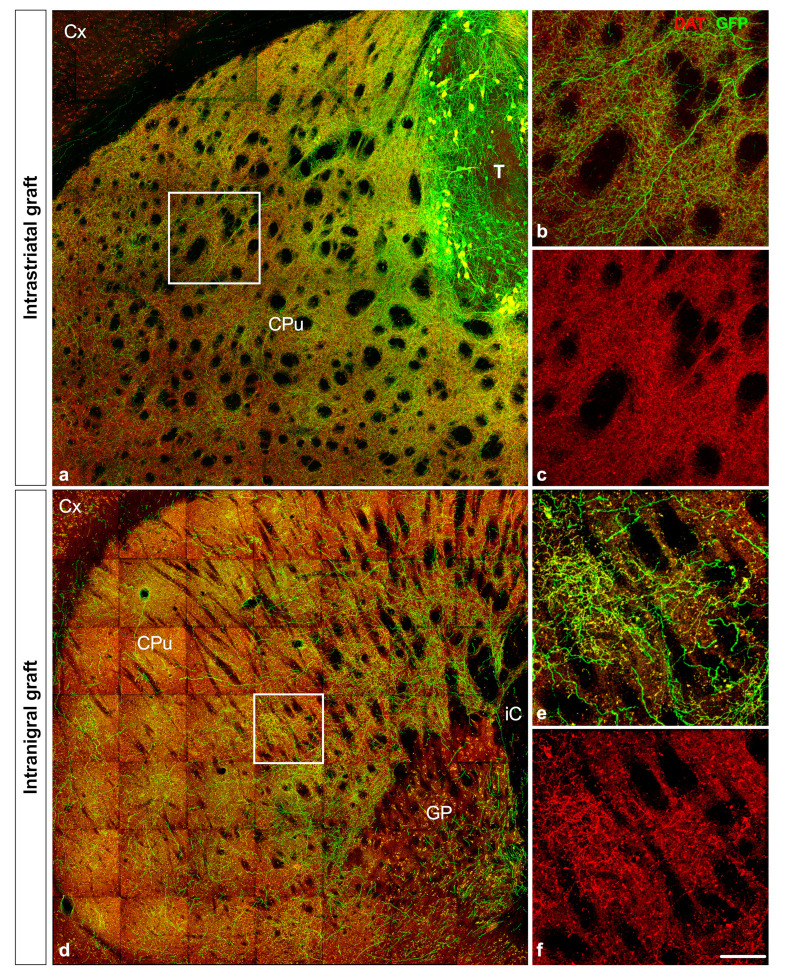
Striatal projections of grafted neurons four months post-grafting. Numerous fibers co-expressed GFP (green) and dopamine transporter (DAT) (red) within the striatum in both the intrastriatal (**a**–**c**) and intranigral (**d**–**f**) conditions. Boxed areas from (**a**,**d**) are shown at higher magnification, respectively, in (**b**,**c**,**e**,**f**) as single-color channels. Scale bar: 60 µm for (**b**,**c**,**e**,**f**), 150 µm for (**a**), and 200 µm for (**d**). Cx: cortex; CPu: caudate putamen; GP: globus pallidus; iC: internal capsule; T: transplant.

**Figure 6 cells-11-01191-f006:**
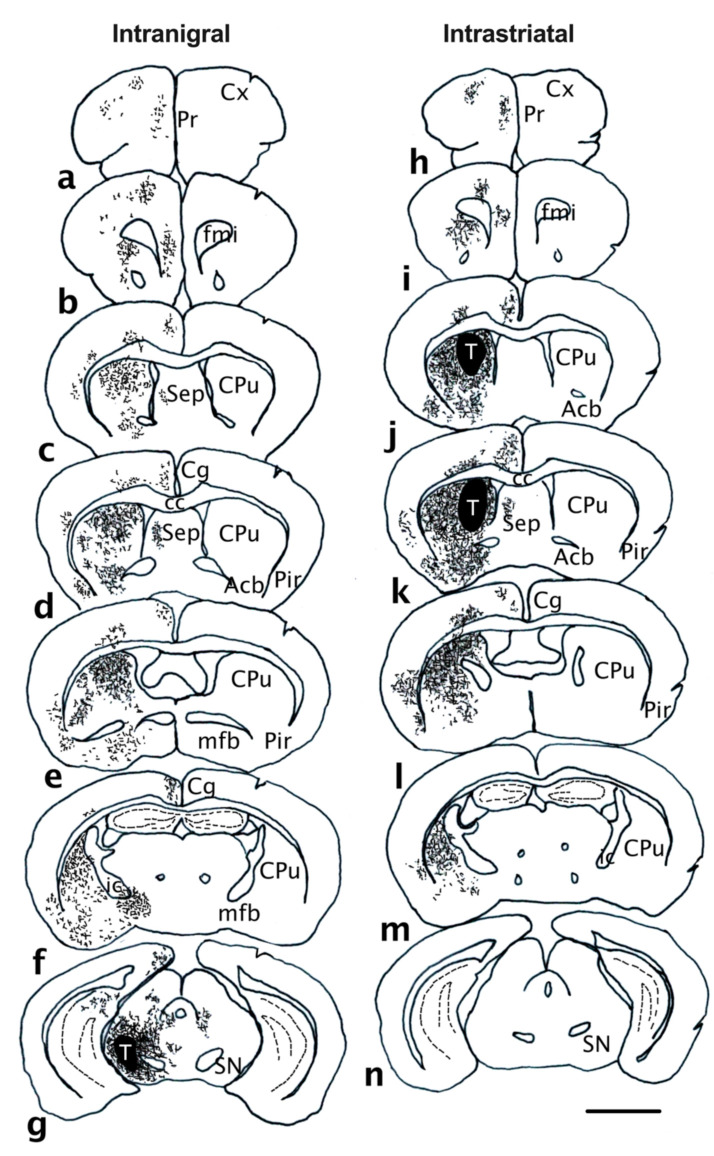
Camera lucida drawings illustrating patterns of dopaminergic fiber outgrowth from intranigral (**a**–**g**) and intrastriatal (**h**–**n**) grafts. Scale bar: 2 mm. Acb: accumbens nucleus; cc: corpus callosum; Cg: cingulate cortex; CPu: caudate putamen; Cx: cortex; ic: internal capsule; fmi: forceps minor of the corpus callosum; mfb: medial forebrain bundle; Pir: perirhinal cortex; Pr: prelimbic cortex; Sep: septum; SN: substantia nigra; T: transplant.

**Figure 7 cells-11-01191-f007:**
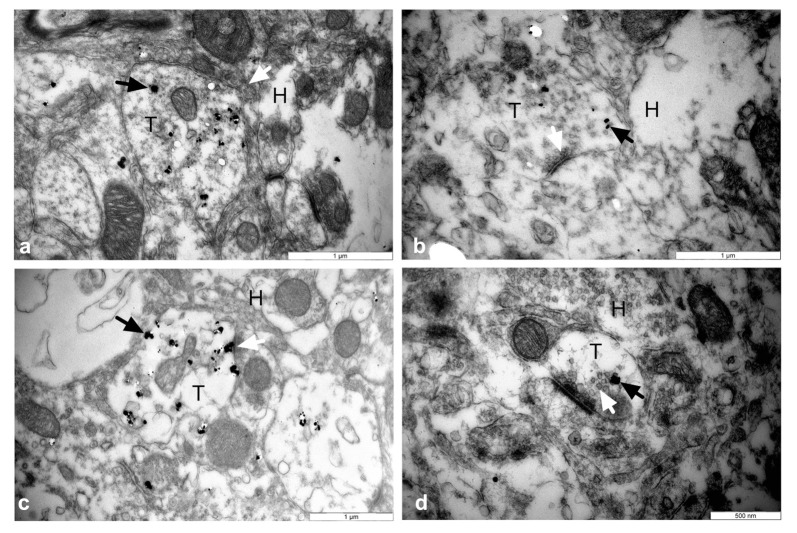
Both intrastriatal- and intranigral-grafted neurons were able to establish reciprocal synaptic contacts with the host circuits. Electron microscopy combined with GFP immunostaining of the synaptic contacts from host to graft within the striatum (**a**) or the substantia nigra (**c**) and from graft to host within the striatum (**b**,**d**) 4 months after grafting. Black arrows show GFP immunogold detection. White arrows show pre-synaptic vesicles. Scale bars represent 1 µm in (**a**–**c**) and 500 nm in (**d**). H: host; T: transplant.

**Figure 8 cells-11-01191-f008:**
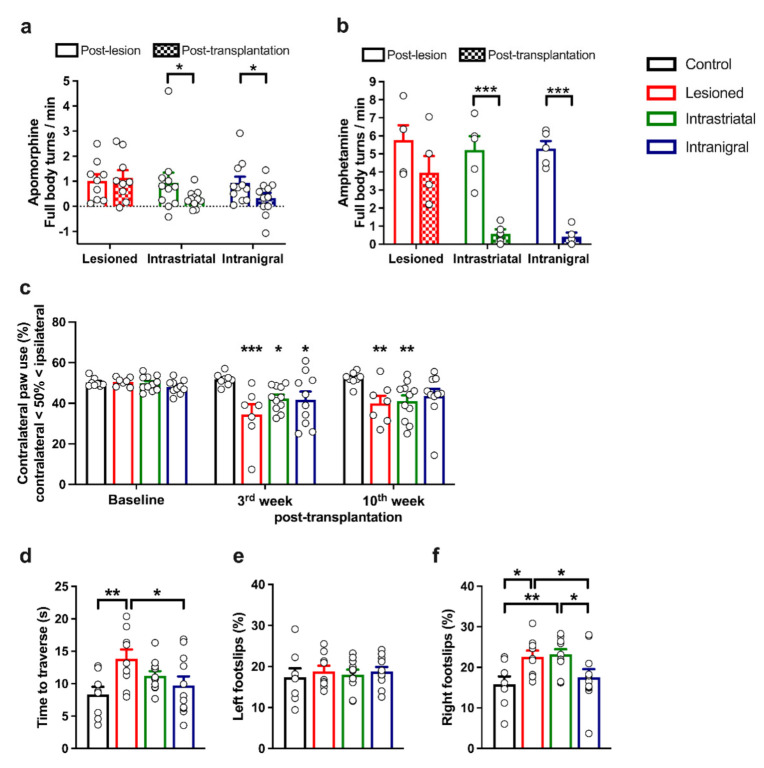
Lesion and transplantation effects on simple behavioral tasks. (**a**) Full body contralateral turns per minute assessed during apomorphine-induced rotation tests. The mice were challenged after the lesioning (post-lesion; empty bars) and after the transplantation (post-transplantation; pattern bars). No difference was found in the lesioned group. Both intrastriatal- and intranigral-grafted mice showed significant reduction in their rotations after the grafting. Data are expressed as mean ± SEM, and asterisks indicate a difference between groups (lesioned, *n* = 9; intrastriatal-grafted, *n* = 11; intranigral-grafted, *n* = 11). (**b**) Full body ipsilateral turns per minute assessed during amphetamine-induced rotation tests; the mice were challenged after lesioning (post-lesion; empty bars) and after the transplantation (post-transplantation; pattern bars). No difference was found in the lesioned group. Both intrastriatal- and intranigral-grafted mice showed significant reduction in their rotations after the grafting. Data are expressed as mean ± SEM, and asterisks indicate a difference between groups (lesioned, *n* = 5; intrastriatal-grafted, *n* = 5; intranigral-grafted, *n* = 5). (**c**) Motor performance for the cylinder test. No difference was found between groups at baseline time. Three weeks following transplantation, all lesioned and grafted mice used their contralateral forepaws significantly fewer times than control mice. Ten weeks post-grafting, while contralateral deficits were still present in both lesioned and intrastriatal-grafted mice, no difference in paw performance was found between control and intranigral-grafted mice (control, *n* = 8; lesioned, *n* = 7; intrastriatal-grafted, *n* = 11; intranigral-grafted, *n* = 10). (**d**–**f**) Motor coordination was assessed using the challenging beam-walking test (control, *n* = 8; lesioned, *n* = 9; intrastriatal-grafted, *n* = 11; intranigral-grafted, *n* = 11). (**d**) Quantification of the time to traverse the beam six weeks after grafting revealed a significant increase in time for the lesioned mice in comparison to both control and intranigral-grafted mice. (**e**) Percentage of foot slips made with the ipsilateral forelimbs; no difference was found between groups. (**f**) Percentage of foot slips made with the contralateral forelimbs. Both lesioned and intrastriatal-grafted mice displayed a significantly greater number of foot slips in comparison to both control and intranigral-grafted mice. (**a**,**b**) Two-way repeated ANOVA measures followed by Fisher’s PLSD. (**c**) Two-way repeated ANOVA measures followed by Dunnett’s multiple comparisons test vs. control. (**d**–**f**) One-way ANOVA followed by Fisher’s PLSD. * *p* < 0.05; ** *p* < 0.01; *** *p* < 0.001.

**Figure 9 cells-11-01191-f009:**
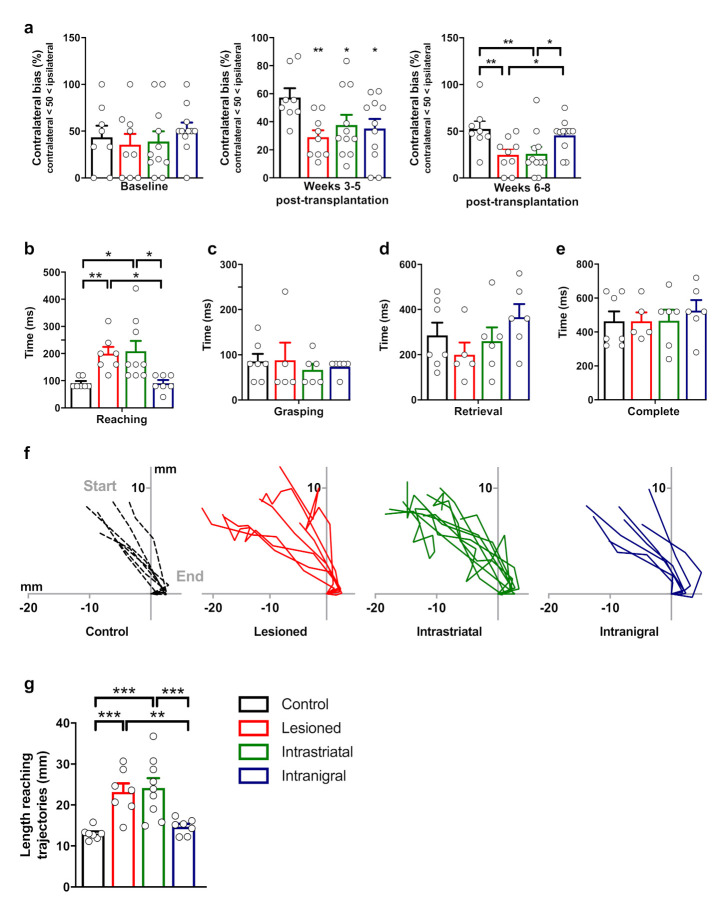
Lesion and transplantation effects on a complex behavioral task. (**a**) Motor performances were assessed using the staircase test. The data were collated over 3 consecutive weeks as baseline and 2 blocks of 3 consecutive weeks post-transplantation (weeks 3–5 and 6–8). At baseline time (left panel), no difference was found between groups. Three to five weeks post-transplantation (middle panel), a reaching deficit was found for all the lesioned and transplanted mice. While this reaching deficit was still present in both the lesioned and intrastriatal mice 6 to 8 weeks post-transplantation (right panel), only intranigral-grafted mice displayed an improvement in reaching performance. (**b**–**e**) Time required to achieve each task phase or the totality of the grasping movement (control, *n* = 8; lesioned, *n* = 9; intrastriatal-grafted, *n* = 11; intranigral-grafted, *n* = 11). (**b**) Both lesioned and intrastriatal-grafted mice required a significantly longer time to achieve the reaching phase in comparison to control and intranigral-grafted mice (control, *n* = 7; lesioned, *n* = 7; intrastriatal-grafted, *n* = 9; intranigral-grafted, *n* = 7). (**c**–**e**) All groups performed the grasping, the retrieval, and the totality of the grasping movement within the same timeframe. (**f**) Lateral view of paw trajectories of the reaching movement corresponding to (**b**). Note that control and intranigral-grafted mice displayed rectilinear trajectories in contrast to the nonlinear outlines of the lesioned and intrastriatal-grafted mice. (**g**) Length of the reaching trajectories corresponding to (**f**). Both lesioned and intrastriatal-grafted mice displayed a significantly longer reaching movement than both control and intranigral-grafted mice (control, *n* = 7; lesioned, *n* = 7; intrastriatal-grafted, *n* = 9; intranigral-grafted, *n* = 7). (**a**,**g**) Comparison between groups was performed using a one-way ANOVA test followed by Fisher’s PLSD and (**b**–**e**) the Kruskal–Wallis test followed by Dunn’s multiple comparisons test. * *p* < 0.05; ** *p* < 0.01; *** *p* < 0.001.

**Figure 10 cells-11-01191-f010:**
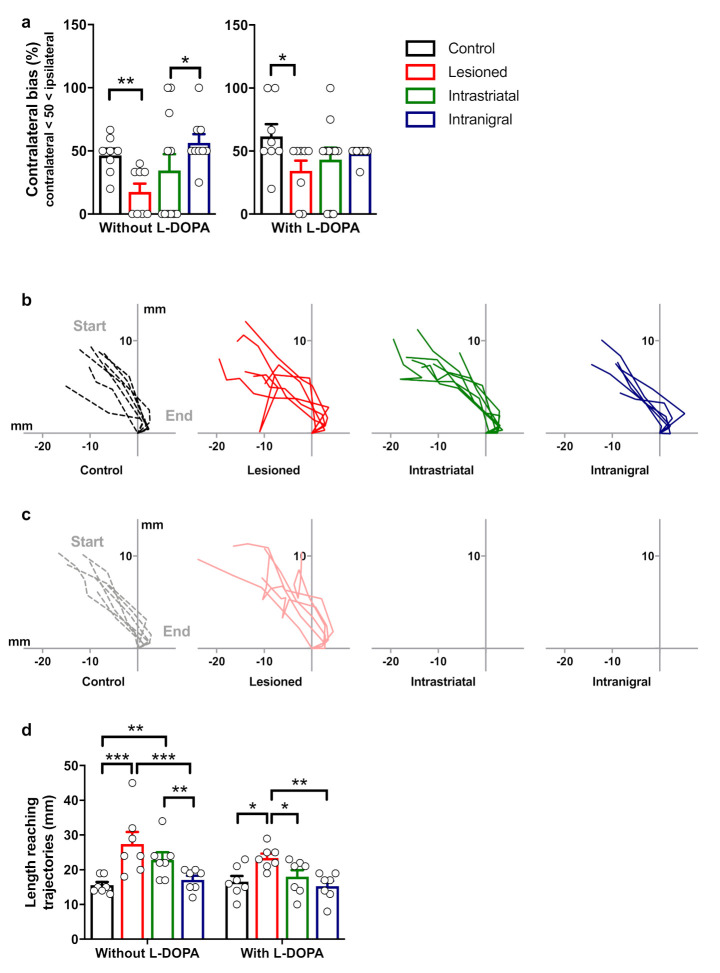
Effects of a single injection of L-DOPA on motor performance. The staircase test was performed 16 weeks post-transplantation during two sessions without and with an L-DOPA injection. (**a**) The data are expressed as the percentage of contralateral bias. Without an L-DOPA injection (left panel), lesioned mice displayed a significant deficit in contralateral paw bias in comparison to both control and intranigral-grafted mice. After an L-DOPA injection (right panel), lesioned mice were significantly different from control mice but not from intranigral-grafted mice (control, *n* = 8; lesioned, *n* = 8; intrastriatal-grafted, *n* = 11; intranigral-grafted, *n* = 9). (**b**) Lateral view of paw trajectories for the reaching movement in the session without L-DOPA injection. Note that, as previously described, both control and intranigral-grafted mice displayed rectilinear trajectories in contrast to the nonlinear outlines of both the lesioned and intrastriatal-grafted mice. (**c**) Lateral view of paw trajectories of the reaching movement in the session after a single L-DOPA injection. (**d**) Length of the reaching trajectories corresponding to (**b**,**c**). Without L-DOPA, both lesioned and intrastriatal-grafted mice exposed longer trajectories than both control and intranigral-grafted mice. In contrast, after a single L-DOPA injection, only the lesioned group exhibited trajectories that were significantly longer than the control and grafted groups (control, *n* = 7; lesioned, *n* = 7; intrastriatal-grafted, *n* = 7; intranigral-grafted, *n* = 7). Comparison between groups in (**a**) was performed using a Kruskal–Wallis test followed by Dunn’s test, and was performed in (**d**) using two-way repeated ANOVA measures followed by Fisher’s PLSD. * *p* < 0.05; ** *p* < 0.01; *** *p* < 0.001.

**Figure 11 cells-11-01191-f011:**
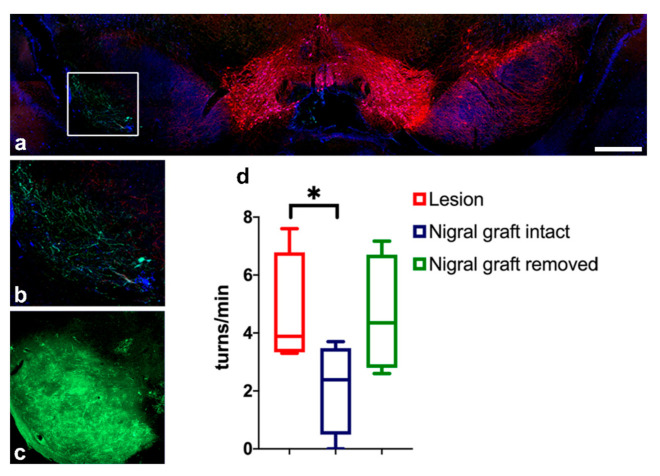
Removal of the grafts abolished graft-induced recovery. (**a**–**c**) Representative images of an intranigral graft after an injection of 6-OHDA into the graft. TH (red), GFP (green), and calbindin (blue) intranigral grafts. (**b**) High-magnification view of boxed area in (**a**) showing an almost complete absence of GFP signal in comparison with an intact intranigral graft (**c**). (**d**) Histogram showing quantifications of amphetamine-induced rotations in lesioned (*n* = 4) animals before intranigral grafting (nigral graft intact; *n* = 4) and after the nigral graft was removed (*n* = 4). Scale bar: 400 μm (**a**,**c**) and 185 μm (**b**). * *p* < 0.05.

**Figure 12 cells-11-01191-f012:**
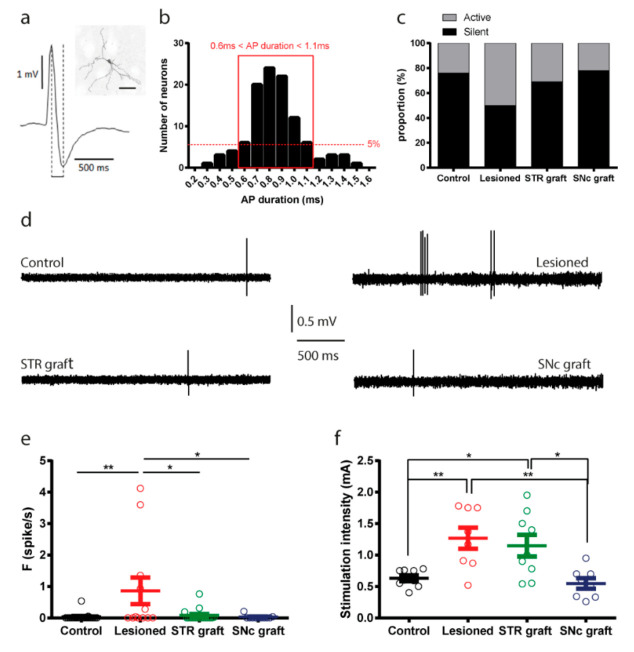
Lesion and transplantation effects on striatal neuron firing properties. (**a**) Example of an action potential and morphology of a medium spiny neuron recorded in the striatum (scale bar = 20 µm). Dashed lines indicate the time points used to measure action potential duration. (**b**) Inclusion criteria of medium spiny neurons based on distribution of action potential durations. The dashed red line indicates the level corresponding to the number of neurons representing 5% of the total neuronal population; the red line indicates the limits of inclusion. (**c**) The proportion of neurons spontaneously active was calculated for each group. (**d**) Examples of spontaneous spike firing recorded in each condition (control, lesioned, STR graft, and SNc graft). Spontaneous spike firing frequency (**e**) and cortico-striatal responses (**f**) were recorded for putative medium spiny neurons in the striatum in control mice (*n* = 14), SNc-lesioned mice (lesioned; *n* = 8), SNc-lesioned and intrastriatal-grafted mice (STR graft; *n* = 8), and SNc-lesioned and intranigral-grafted mice (SNc graft; *n* = 8). * *p* < 0.05; ** *p* < 0.01.

**Table 1 cells-11-01191-t001:** Summary of the number of animals used in the different groups.

Total 118	112 Remaining	22 Intact			
		90 Lesioned	28 Lesioned		
			62 Transplanted	6 β-actin-GFP mouse fetal VM tissue	3 Intrastriatal
					3 Intranigral
				56 TH-GFP mouse fetal VM tissue	26 Intrastriatal
					30 Intranigral
	6 Excluded (Failed to reach the criterion for the staircase test training period)				

**Table 2 cells-11-01191-t002:** A summary table of the results obtained.

		Intrastriatal	Vs	Intranigral
Anatomical comparison	Transplants	Graft survival		
		Decrease in GFP+ neurons within the graft		
	Expression of dopaminergic markers	Same proportion of GFP/Calb+ neurons Same proportion of GFP/Girk2+ neurons		
	Differentiation into mature neurons		Increase in GFP/NeuN+ cells	
	Astrocytic evaluation	Increase in GFAP+ cells (surrounding and within the graft)		
	Level of inflammation	Increase in Iba1+ cells (surrounding and within the graft)		
	Inflammatory profile	Increase of Iba1/CD68+ cells	Increase of Iba1/Arg1+ cells	
		Increase in GFAP/C3+ cells (surrounding and within the graft)		
		Low percentage of GFAP/CD109 + cells		
	Projection of grafted neurons	Send projections to appropriate dopaminergic targets		
	Pathway taken by projections	Covering the entire rostrocaudal extent of the striatum Present in the Acb, FC and PC	Nigrostriatal and mesolimbocortical pathway Extend through the mfb, nigrostriatal pathway, GP exiting the dorsal striatum	
Functional comparison	Synaptic contacts	Establish reciprocal synaptic contacts with the host circuits		
	Rotation tests	Decrease apomorphine- or amphetamine-induced rotations		
	Simple and complex behavioral tasks	No improvement in all behavioral tests	Improvement in all behavioral tests	
	L-DOPA administration	L-DOPA improved motor performance	No potentiation with L-DOPA	
	Electrophysiological assessment	Restoration of excitability		
		Increase cortical stimulation threshold	Normalize cortico- striatal responses	

Acb: Accumbens nucleus; Calb: Calbindin; FC: frontal cortex; GFAP: glial fibrillary acidic protein; Girk2: G-protein-coupled in wardly rectifying potassium channels; GP: globus pallidus; Mfb: medial forebrain bundle; PC: perirhinal cortex.

## Data Availability

Some or all data used during the study are available from the corresponding author by request.

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
