# Peer review of "Better Outcomes with Intranigral versus Intrastriatal Cell Transplantation: Relevance for Parkinson’s Disease"

_cells, 2022, doi:10.3390/cells11071191_

Round 1
Reviewer 1 Report
This manuscript is aimed at performing a comparison of the putative benefits exerted by intranigral or intrastriatal graft in an experimental model of Parkinson's disease. The collected results show interesting outcomes regarding both intranigral and intrastriatal grafts, with intranigral being the most effective approach. The manuscript is well written and the results adequately discussed and integrated in the state of the art.
However, some shortcomings should be addressed:
- Results section, paragraph 3.1: the reported results are not properly referred to Figures/Tables. If these data are not shown, please include them in a revised version of the manuscript, at least as "supplementary data".
- Figure 1: panels for Calbindin and Girk appears with low resolution.
- The number of replicates should be clearly indicated for each experiment in the respective figure legends
- Figure 2: figure panels are not numbered with letters, thus the interpretation of the image is not easy. Please fix this inconsistency.
- It is not clear which kind of "control" animals are used in this study. Notably, two kind of control groups should be included: control naive "untouched" animals, and animals underwent sham procedures (mimicking, for instance, 6-OHDA injection).
Author Response
We thank this Reviewer for his/her support and most useful criticisms, which we have tried to address in full, as detailed below.
Results section, paragraph 3.1: the reported results are not properly referred to Figures/Tables. If these data are not shown, please include them in a revised version of the manuscript, at least as "supplementary data".
The data presented in section 3.1 are related to the degree of dopaminergic lesion and the location of the grafts ...We didn't illustrate the results concerning the degree of the lesion, as this data is extensively documented and published by us and other teams (ref. 37). Data concerning the location of the grafts and their content are illustrated in figures 2, 5 and 6.
Figure 1: panels for Calbindin and Girk appears with low resolution.
Figure 1 has been replaced by a tiff format (600 dpi).
The number of replicates should be clearly indicated for each experiment in the respective figure legends
This has been corrected
Figure 2: figure panels are not numbered with letters, thus the interpretation of the image is not easy. Please fix this inconsistency.
The information concerning this figure disappeared accidentally when placed within the article in Word format. The problem is now solved, and the image has been replaced.
It is not clear which kind of "control" animals are used in this study. Notably, two kind of control groups should be included: control naive "untouched" animals, and animals underwent sham procedures (mimicking, for instance, 6-OHDA injection).
Control animals used are intact and only injured mice. These control animals are mainly used for functional studies.
Reviewer 2 Report
This manuscript reports a very extensive and thorough comparison of striatal vs nigral implantation of embryonic ventral midbrain cells in 6-OHDA lesioned mice. This therapeutic approach has been investigated for more than 2 decades, and still remains a promising approach through various advances and optimizations, including the use of iPS derived dopaminergic neurons. Several targets of implantation have been evaluated during this time, to determine the most appropriate site. In this report, the authors performed a comprehensive evaluation of multiple outcomes following long-term recovery from nigral or striatal implants of ventral midbrain tissue. This is a very detailed and comprehensive work, showing remarkable functional and pathological improvement in the 6-OHDA model, and it remains an area of high interest to the PD community. There are a few points that need addressing before publication, as outlined below.
- In section 2.1 of the Methods, the description of the different animal groups would be better presented in a Table format.
- What was the rationale for the unilateral 6-OHDA injections in the nigra and not the striatum?
- In the transplantation section of the Methods, there are references to GFP and EGFP transgenic mice. I suspect this is a typo, but clarify if this refers to the same cDNA in both lines?
- More details in the transplantation section are needed. Specifically, were the nigral or striatal injections unilateral or bilateral? If unilateral, in which side were the cells transplanted? What % of TH-positive neurons were routinely obtained under the conditions described? What was the total number of cells implanted? It is only written that the density of cells was 150,000/ul, not how many ul were injected. In this same section (2.2), line 132, it is written that the implantations were performed in an alternating manner, it is not clear what this is referring to, or why it was done this way?
- In Section 3.2, line 413, the authors conclude that more cells transplanted into the striatum actively differentiated into astrocytes compared to cells implanted into the nigra. What is the basis of this conclusion, it is not clear which data reported in the manuscript support this conclusion of altered differentiation of cells in the different implant regions.
- Section 3.3 in the Results, the plots of the Iba1 data, shown in Figure 4 should be shown
- Did the authors examine other markers of inflammation (e.g. circulating pro-inflammatory mediators)?
- Given the broad range of experiments performed, perhaps a good way to summarize the findings, in a high level way, would be in a Table or schematic.
- One important area that merits discussion in the context of the present findings is an integration of the findings from several years ago of reports showing transmission of host-graft synuclein aggregates. This remains a critical concern, uniquely affecting this therapeutic approach. While the 6-OHDA model is not a model of PD accompanied by a-synuclein aggregation, however the authors should comment on this, and specifically what are the implications, if any, with regards to the choice of implant region.
Author Response
We thank this Reviewer for his/her support and criticisms, which we have addressed as detailed below.
The one thing missing in the discussion is the clinically relevant point of difference between target sizes for striatal and nigral administration; I would suggest including this into review of clinical implications.
Because of its location deep in the brain and its size, transplantation into the substantia nigra is more challenging than transplantation into the striatum.
Minor (very minor) issues:
On line 165 – “turn were”
This has been corrected
Refs 58 and 63 are incomplete or incorrect
This has been corrected
Reviewer 3 Report
The paper is very good – although perhaps too long and too comprehensive; normally, I would suggest splitting it into several separate papers.
The one thing missing in the discussion is the clinically relevant point of difference between target sizes for striatal and nigral administration; I would suggest including this into review of clinical implications.
Minor (very minor) issues:
On line 165 – “turn were”
Refs 58 and 63 are incomplete or incorrect
Author Response
We thank this Reviewer for his/her support and most useful criticisms, which we have tried to address in full, as detailed below.
In section 3.3 microglia polarization. The M1 and M2 nomenclature has been superseded in recent years. Researchers now generally use increase in CD68 and loss of homeostasis marker P2RY12 to report activation of ‘neurotoxic’ microglia (See Schulz et al. 2016). The data shown in Droguerre et al. is solid and well reported therefore I would only suggest to rephrase the text (results and discussion) to update and avoid old nomenclature for readers that may not be aware of the recent advances in the microglia field.
We did not find the article by Schulz et al. 2016, but we have replaced the terms M1 and M2 in the manuscript respectively by neurotoxic and neuroprotective phenotype.
Figure 6 is a very useful figure and clearly depicts one of the main messages. Perhaps I missed this, but could the authors explain how this data was generated and perhaps include some of the original images used to generate the figure in a supplementary file if possible?
The images are generated using a camera lucida which is an optical device attached to the eyepiece of a microscope, through which an image of the brain section is projected onto a sheet of paper for tracing.
The EM data and the electrophysiology data is very good. While is acceptable that the EM data provides a qualitative assessment, the electrophysiology/ activity data is quantified. Are some example traces from each condition group available to illustrate this work better? I only see one example trace in Figure 12A.
In order to illustrate better the electrophysiological study, Figure 12 have been modified and example of traces showing spontaneous spike firing have been added for each experimental group (Figure 12 d).
Again, perhaps I missed this, but I think it is worth noting more strongly in the discussion that the activity of the neurons is one outcome that seems to be not significantly improved or worsened by intranigral versus intrastriatal transplant.
The electrophysiological results are now discussed (p28) and the reviewer’s point is addressed in this paragraph.
Reviewer 4 Report
Droguerre and colleagues have provided an interesting and excellent study in mice showing that intranigral cell transplantation may be beneficial over intrastriatal transplantation for the treatment of Parkinson’s disease.
The key benefits include reduced inflammation and better outcome in terms of improved fine motor control.
The manuscript is well written and the data, figures and methods are clearly presented.
I only have some minor comments/suggestions:
- In section 3.3 microglia polarization. The M1 and M2 nomenclature has been superseded in recent years. Researchers now generally use increase in CD68 and loss of homeostasis marker P2RY12 to report activation of ‘neurotoxic’ microglia (See Schulz et al. 2016). The data shown in Droguerre et al. is solid and well reported therefore I would only suggest to rephrase the text (results and discussion) to update and avoid old nomenclature for readers that may not be aware of the recent advances in the microglia field.
- Figure 6 is a very useful figure and clearly depicts one of the main messages. Perhaps I missed this, but could the authors explain how this data was generated and perhaps include some of the original images used to generate the figure in a supplementary file if possible?
- The EM data and the electrophysiology data is very good. While is acceptable that the EM data provides a qualitative assessment, the electrophysiology/ activity data is quantified. Are some example traces from each condition group available to illustrate this work better? I only see one example trace in Figure 12A. Again, perhaps I missed this, but I think it is worth noting more strongly in the discussion that the activity of the neurons is one outcome that seems to be not significantly improved or worsened by intranigral versus intrastriatal transplant.
Author Response
We thank this Reviewer for his/her support and most useful criticisms, which we have tried to address in full, as detailed below.
In section 2.1 of the Methods, the description of the different animal groups would be better presented in a Table format.
Summary of the number of animals used in different groups is now included in the method section
What was the rationale for the unilateral 6-OHDA injections in the nigra and not the striatum?
A single injection of 6-OHDA at the level of the SN induces an important dopaminergic lesion with measurable motor deficits. A single injection of 6-OHDA at the striatum level does not allow to obtain important dopaminergic lesions and, it is necessary to perform several injections at different anteroposterior levels of the striatum in order to obtain important lesions.
In the transplantation section of the Methods, there are references to GFP and EGFP transgenic mice. I suspect this is a typo, but clarify if this refers to the same cDNA in both lines?
This has now been corrected
More details in the transplantation section are needed. Specifically, were the nigral or striatal injections unilateral or bilateral? If unilateral, in which side were the cells transplanted?
Substantia nigra or the striatum of 6-OHDA-lesioned mice were transplanted unilaterally at the following stereotaxic coordinates (respectively: AP, −3.1; ML, 1.3; DV, 3.7 and AP, 0.5; ML, 2; DV, 2.8). This has been now specified in the text.
What % of TH-positive neurons were routinely obtained under the conditions described?
At the time of transplantation, the number of dopaminergic neurons in the cell suspension prepared for transplantation is not known, as these cells are at a precursor stage. It is well documented that about 90–95% of transplanted neurons fail to survive with most dying by apoptotic cell death (Kordower et al., 1995, Freed et al., 2001). The survival rate of TH neurons within the transplant is about 5 to 10%.
What was the total number of cells implanted? It is only written that the density of cells was 150,000/ul, not how many ul were injected.
This has been now specified in the text.
In this same section (2.2), line 132, it is written that the implantations were performed in an alternating manner, it is not clear what this is referring to, or why it was done this way?
To account for possible cell degradation during the transplantation procedure and that may affect the outcome between the two groups (intranigral vs intrastriatal), we alternated animals from each group during transplantation.
In Section 3.2, line 413, the authors conclude that more cells transplanted into the striatum actively differentiated into astrocytes compared to cells implanted into the nigra. What is the basis of this conclusion, it is not clear which data reported in the manuscript support this conclusion of altered differentiation of cells in the different implant regions.
The conclusion is drawn from results presented in Figure 2. These results are illustrated in Figure 2 h-m and the quantification of the results as a histogram are in Figure 2n.
Section 3.3 in the Results, the plots of the Iba1 data, shown in Figure 4 should be shown
The results of the quantification in Figure 4 are given in the text of the manuscript. As we have many figures in the manuscript, we did not add the results of the quantification as a histogram.
Did the authors examine other markers of inflammation (e.g. circulating pro-inflammatory mediators)?
We focused only on central neuroinflammatory process, we did not analyze the circulating pro-inflammatory mediators.
Given the broad range of experiments performed, perhaps a good way to summarize the findings, in a high level way, would be in a Table or schematic.
Summary of the results are presented as a table in the manuscript.
One important area that merits discussion in the context of the present findings is an integration of the findings from several years ago of reports showing transmission of host- graft synuclein aggregates. This remains a critical concern, uniquely affecting this therapeutic approach. While the 6-OHDA model is not a model of PD accompanied by a-synuclein aggregation, however the authors should comment on this, and specifically what are the implications, if any, with regards to the choice of implant region.
This reviewer is probably referring to three back to back publications from different groups in Nature Medicine in 2008 detailing long term outcome of transplanted cells in Parkinson’s disease patients [Mendez, I. et al. (2008); Kordower, J.H. et al. (2008) and Li, J.Y. et al. (2008). Two of these papers [Kordower, J.H. et al. (2008) and Li, J.Y. et al. (2008)] reported that a very small percentage (1 to 5%) of transplanted cells contained cytoplasmic inclusions that resembled to Lewy body (LB) aggregates. The authors concluded that PD pathology may, in part, be transmitted to the newly transplanted cells. It has been suggested that the presence of LBs in the transplanted cells may be the result of ectopic placement of fetal VM cells within the caudate putamen where they lack normal SN local neural growth-factor support (Mendez, I. et al., 2008; Cooper, O. et al., 2009). This has been extensively discussed in our review in TINS and that is referenced in this manuscript [Gaillard, A.; Jaber, M. Rewiring the brain with cell transplantation in Parkinson's disease. Trends Neurosci 2011, 34, 124-133]. We have also discussed this point in the manuscript.
Round 2
Reviewer 1 Report
The authors properly addressed the reviewer's concerns by providing a clear point-by-point response letter.